# nGPT: Normalized Transformer with Representation Learning on the Hypersphere

**Ilya Loshchilov, Cheng-Ping Hsieh, Simeng Sun & Boris Ginsburg**
NVIDIA
{iloshchilov,chsieh,simengs,bginsburg}@nvidia.com

## Abstract

We propose a novel neural network architecture, the normalized Transformer (nGPT) with representation learning on the hypersphere. In nGPT, all vectors forming the embeddings, MLP, attention matrices and hidden states are unit norm normalized. The input stream of tokens travels on the surface of a hypersphere, with each layer contributing a displacement towards the target output predictions. These displacements are defined by the MLP and attention blocks, whose vector components also reside on the same hypersphere. Experiments show that nGPT learns much faster, reducing the number of training steps required to achieve the same accuracy by a factor of 4 to 20, depending on the sequence length.

## 1 Introduction

The Transformer architecture (Vaswani et al., 2017) is the foundation for most of modern language models. An enormous number of modifications to this architecture have been proposed to improve training stability, inference costs, context length, robustness, etc. It has been noted that the application of various normalization techniques is beneficial (Salimans & Kingma, 2016), leading to experiments with adding normalization layers such as LayerNorm and RMSNorm in nearly every possible position within the network (Xiong et al., 2020). Another approach to the model normalization is through controlling the norm of weights using weight decay (Loshchilov & Hutter, 2019). Recent studies (Andriushchenko et al., 2023) suggest reevaluating the role of weight decay and taking a closer look at rotations rather than focusing solely on vector norms (Kodryan et al., 2022; Kosson et al., 2023). Franke et al. (2023) suggested to enforce an upper bound on $L_2$ norm of parameter groups. There is growing evidence that representation learning on the hypersphere is associated with more stable training, greater embedding space separability, and better performance on downstream tasks (Wang & Isola, 2020). Recent studies also suggest that transformers implicitly perform gradient descent as meta-optimizers (Von Oswald et al., 2023; Dai et al., 2022).

We propose to unify (see also Appendix A.3) various findings and observations made in the field under a new perspective of the normalized Transformer. Our key contributions are as follows:

**Optimization of network parameters on the hypersphere** We propose to normalize all vectors forming the embedding dimensions of network matrices to lie on a unit norm hypersphere. This allows us to view matrix-vector multiplications as dot products representing cosine similarities bounded in [-1,1]. The normalization renders weight decay unnecessary.

**Normalized Transformer as a variable-metric optimizer on the hypersphere** The normalized Transformer itself performs a multi-step optimization (two steps per layer) on a hypersphere, where each step of the attention and MLP updates is controlled by eigen learning rates—the diagonal elements of a learnable variable-metric matrix. For each token $t_i$ in the input sequence, the optimization path of the normalized Transformer begins at a point on the hypersphere corresponding to its input embedding vector and moves to a point on the hypersphere that best predicts the embedding vector of the next token $t_{i+1}$.

**Faster convergence** We demonstrate that the normalized Transformer (nGPT) reduces the number of training steps required to achieve the same accuracy by a factor of 4 to 20.

## 2 EVOLUTION OF THE TRANSFORMER: FROM GPT TO NGPT

This section outlines the baseline Transformer and the modifications necessary to derive its normalized version. We illustrate these changes for Transformer decoder with self-attention only. The extension to encoder-decoder and cross-attention is straightforward. A summary of these changes is given in Table 1, with details in Section 2.6.

### 2.1 TOKEN EMBEDDINGS AND OUTPUT LOGITS

The decoder-only Transformer is trained to predict token $t_i$ using previous tokens input sequence $\boldsymbol{x} = (t_1, t_2, \ldots, t_{i-1})$. For each input token $t_i$, we retrieve its corresponding embedding in $\mathbb{R}^{d_{\text{model}}}$ from a learnable embedding matrix $\boldsymbol{E}_{\text{input}} \in \mathbb{R}^{V \times d_{\text{model}}}$ with vocabulary of size $V$. Similarly, the target output sequence $\boldsymbol{y}$ is represented using a learnable embedding matrix $\boldsymbol{E}_{\text{output}} \in \mathbb{R}^{V \times d_{\text{model}}}$. Notably, since both $\boldsymbol{E}_{\text{input}}$ and $\boldsymbol{E}_{\text{output}}$ are learnable (unless they are tied to be equivalent), any token $t_i$ can have different embeddings in the input and output sequences. To measure token similarity during model training, the dot product between the corresponding embedding vectors is used. However, the norms of embedding vectors in the original Transformer are unconstrained, which can lead to inaccurate similarity estimation. To improve the accuracy of similarity estimation, we propose to normalize the embedding vectors stored in $\boldsymbol{E}_{\text{input}}$ and $\boldsymbol{E}_{\text{output}}$ after each step of the training algorithm.

The next token prediction is enforced by causal masking (see Section 2.3) to ensure that no future tokens are considered. This allows the model to compute the prediction error for all $T$ tokens in parallel during training, while preserving the autoregressive nature of the task. After the Transformer processes the sequence $(\boldsymbol{x}_1, \ldots, \boldsymbol{x}_{i-1})$, it produces an output vector $\boldsymbol{h}_i \in \mathbb{R}^{d_{\text{model}}}$ for each $i$-th position in the predicted sequence. The logits $\boldsymbol{z}_i \in \mathbb{R}^V$, representing the unnormalized probabilities for each token in the vocabulary, are computed using the output embedding matrix $\boldsymbol{E}_{\text{output}}$:

$$\boldsymbol{z}_i = \boldsymbol{E}_{\text{output}} \boldsymbol{h}_i \tag{1}$$

The logits $\boldsymbol{z}_i$ are passed through a softmax function to convert them into probabilities:

$$P(y_i | \boldsymbol{x}_1, \ldots, \boldsymbol{x}_{i-1}) = \frac{\exp(z_{i,y_i})}{\sum_{v=1}^{V} \exp(z_{i,v})} \tag{2}$$

Here, $z_{i,y_i}$ is the logit corresponding to the correct token $y_i$, and the denominator normalizes the logits into a probability distribution over the vocabulary. During inference, the prediction $\hat{y}_i$ is obtained by selecting the token with the highest probability. Since all nGPT embeddings are normalized, the logits $\boldsymbol{z} \in \mathbb{R}^V$ in equation 1 represent dot products bounded in the range $[-1, 1]$. This limits the confidence (temperature) of the probability distribution generated by the softmax in equation 2. To adjust this during training, and in line with Hoffer et al. (2018), we introduce a trainable scaling parameter $\boldsymbol{s}_z \in \mathbb{R}^V$ that scales the logits element-wise:

$$\boldsymbol{z} \leftarrow \boldsymbol{z} \boldsymbol{s}_z \tag{3}$$

Table 1: Transformer vs. Normalized Transformer.

| Transformer | Normalized Transformer |
|---|---|
| $\boldsymbol{h}_{\text{A}} \leftarrow \text{ATTN}(\text{RMSNorm}(\boldsymbol{h}))$ | $\boldsymbol{h}_{\text{A}} \leftarrow \text{Norm}(\text{ATTN}(\boldsymbol{h}))$ |
| $\boldsymbol{h} \leftarrow \boldsymbol{h} + \boldsymbol{h}_{\text{A}}$ | $\boldsymbol{h} \leftarrow \text{Norm}(\boldsymbol{h} + \boldsymbol{\alpha}_{\text{A}}(\boldsymbol{h}_{\text{A}} - \boldsymbol{h}))$ |
| $\boldsymbol{h}_{\text{M}} \leftarrow \text{MLP}(\text{RMSNorm}(\boldsymbol{h}))$ | $\boldsymbol{h}_{\text{M}} \leftarrow \text{Norm}(\text{MLP}(\boldsymbol{h}))$ |
| $\boldsymbol{h} \leftarrow \boldsymbol{h} + \boldsymbol{h}_{\text{M}}$ | $\boldsymbol{h} \leftarrow \text{Norm}(\boldsymbol{h} + \boldsymbol{\alpha}_{\text{M}}(\boldsymbol{h}_{\text{M}} - \boldsymbol{h}))$ |
| Final: $\boldsymbol{h} \leftarrow \text{RMSNorm}(\boldsymbol{h})$ | |
| All parameters of matrices and embeddings are unconstrained. | After each batch pass, all matrices and embeddings are normalized along their embedding dimension. The hidden state updates are controlled by learnable vectors of eigen learning rates $\boldsymbol{\alpha}_{\text{A}}$ and $\boldsymbol{\alpha}_{\text{M}}$. |

## 2.2 LAYERS AND BLOCKS

### 2.2.1 BASELINE TRANSFORMER

$L$ layers of transformations are applied to the hidden state $\boldsymbol{h}$, consisting of alternating the self-attention (ATTN) and multi-layer perceptron (MLP) blocks:

$$\boldsymbol{h} \leftarrow \boldsymbol{h} + \text{ATTN}(\text{RMSNorm}(\boldsymbol{h})) \tag{4}$$

$$\boldsymbol{h} \leftarrow \boldsymbol{h} + \text{MLP}(\text{RMSNorm}(\boldsymbol{h})), \tag{5}$$

where $\text{RMSNorm}(\boldsymbol{h})$ is one of several possible normalizations. It is used first to normalize each embedding to a norm of $\sqrt{d_{\text{model}}}$, then scales each dimension by a learnable vector of $d_{\text{model}}$ factors, typically initialized to 1. Since the transformation block outputs are added to $\boldsymbol{h}$, the token embedding norms can vary significantly. To address this, normalization is also applied after the final layer.

### 2.2.2 NORMALIZED TRANSFORMER

For any points $\boldsymbol{a}$ and $\boldsymbol{b}$ on the surface of a hypersphere in $\mathbb{R}^{d_{\text{model}}}$, SLERP (Spherical Linear Interpolation) by Shoemake (1985) computes an interpolation along the geodesic (shortest path):

$$\text{SLERP}(\boldsymbol{a}, \boldsymbol{b}; \alpha) = \frac{\sin((1-\alpha)\theta)}{\sin(\theta)}\boldsymbol{a} + \frac{\sin(\alpha\theta)}{\sin(\theta)}\boldsymbol{b} \tag{6}$$

where $\theta = \arccos(\boldsymbol{a} \cdot \boldsymbol{b})$ is the angle between the points $\boldsymbol{a}$ and $\boldsymbol{b}$, and $\alpha \in [0,1]$ is the interpolation parameter, with $\alpha = 0$ returning $\boldsymbol{a}$ and $\alpha = 1$ returning $\boldsymbol{b}$. Our experiments suggest that SLERP can be approximated by simple linear interpolation (LERP):

$$\text{LERP}(\boldsymbol{a}, \boldsymbol{b}; w) = (1-\alpha)\boldsymbol{a} + \alpha\boldsymbol{b} \tag{7}$$

Let us rewrite this equation as an update equation in nGPT:

$$\boldsymbol{a} \leftarrow \boldsymbol{a} + \alpha(\boldsymbol{b} - \boldsymbol{a}) \tag{8}$$

where $\boldsymbol{a}$ is $\boldsymbol{h}$, and, $\boldsymbol{b}$ is the point suggested by the attention or MLP block. Then, for the gradient $\boldsymbol{g} = \boldsymbol{a} - \boldsymbol{b}$, a more general form involving a variable matrix $\boldsymbol{B} \in \mathbb{R}^{d_{\text{model}} \times d_{\text{model}}}$ becomes:

$$\boldsymbol{a} \leftarrow \boldsymbol{a} - \alpha\boldsymbol{B}\boldsymbol{g} \tag{9}$$

In quasi-Newton methods, $\boldsymbol{B}$ approximates the inverse Hessian matrix $\boldsymbol{H}^{-1}$. When $\boldsymbol{B}$ is diagonal with non-negative elements, $\alpha\boldsymbol{B}$ becomes a vector $\boldsymbol{\alpha} \in \mathbb{R}^{d_{\text{model}}}_{\geq 0}$ whose elements correspond to the diagonal of $\boldsymbol{B}$ times the learning rate $\alpha$. We denote $\boldsymbol{\alpha}$ as **eigen** learning rates (from the German word eigen, meaning "own," referring to the internal structure of the Transformer). We provide some notes in Appendix A.2. Following equation 8, the update equations for the attention and MLP blocks are as follows:

$$\boldsymbol{h} \leftarrow \text{Norm}(\boldsymbol{h} + \boldsymbol{\alpha}_{\text{A}}(\boldsymbol{h}_{\text{A}} - \boldsymbol{h})) \tag{10}$$

$$\boldsymbol{h} \leftarrow \text{Norm}(\boldsymbol{h} + \boldsymbol{\alpha}_{\text{M}}(\boldsymbol{h}_{\text{M}} - \boldsymbol{h})), \tag{11}$$

where $\boldsymbol{\alpha}_{\text{A}} \in \mathbb{R}^{d_{\text{model}}}_{\geq 0}$ and $\boldsymbol{\alpha}_{\text{M}} \in \mathbb{R}^{d_{\text{model}}}_{\geq 0}$ are learnable parameters applied to the normalized outputs of the attention and MLP blocks $\boldsymbol{h}_{\text{A}} = \text{Norm}(\text{ATTN}(\boldsymbol{h}))$ and $\boldsymbol{h}_{\text{M}} = \text{Norm}(\text{MLP}(\boldsymbol{h}))$, respectively. The function $\text{Norm}(\boldsymbol{x})$ normalizes any vector $\boldsymbol{x}$ to have unit norm, and, unlike RMSNorm or LayerNorm, does not introduce any element-wise scaling factors. The normalization can be viewed as the retraction step in Riemannian optimization, mapping the updated solution back to the manifold. Appendix A.4 discusses the extension of our update equations in the context Riemannian optimization. In contrast to the baseline Transformer, no additional normalization is required after the final layer, as the embeddings are already normalized by the proposed scheme.

## 2.3 SELF-ATTENTION BLOCK

### 2.3.1 BASELINE TRANSFORMER

The attention mechanism is a key component of the Transformer. It allows each token to attend to every other token in the sequence, enabling the model to capture long-range dependencies. The

block typically starts with a normalization of the input hidden state $h$ using RMSNorm to deal with fluctuating norms of embeddings. Then, the normalized $h$ is projected into three separate vectors - the query $q$, the key $k$, and the value $v$:

$$q \leftarrow hW_q, k \leftarrow hW_k, v \leftarrow hW_v \tag{12}$$

where $W_q, W_k, W_v \in \mathbb{R}^{d_{\text{model}} \times d_k}$ are learned projection matrices, and $d_k$ is the dimensionality of the query/key vectors. To incorporate positional information, we apply Rotary Position Embeddings (RoPE) by Su et al. (2024) to both the query and key vectors. The attention scores are computed by taking the dot product of the query and key vectors, scaling them by $\frac{1}{\sqrt{d_k}}$, then applying a softmax function to obtain attention weights, and finally computing a weighted sum of the value vectors $v$:

$$\text{Attention}(q, k, v) \leftarrow \text{softmax}\left(\frac{qk^\top}{\sqrt{d_k}} + M\right) v, \tag{13}$$

where $M$ is a matrix that prevents attending to future tokens by setting the corresponding entries to $-\infty$. Specifically, $M_{i,j} = 0$ if $j \leq i$ and $M_{i,j} = -\infty$ if $j > i$.

In practice, $n_{\text{heads}}$ attention heads are used where for each $i$-th head, separate linear projections $W_q^i, W_k^i, W_v^i$ are applied, and the attention mechanism is computed independently for each head:

$$h_A \leftarrow \text{Concat}(\text{head}_1, \dots, \text{head}_{n_{\text{heads}}})W_o \tag{14}$$

where $\text{head}_i = \text{Attention}(q^i, k^i, v^i)$ and $W_O \in \mathbb{R}^{n_{\text{heads}} \times d_k \times d_{\text{model}}}$ is a learned projection matrix, where $d_k$ is typically set to $d_{\text{model}}/n_{\text{heads}}$.

### 2.3.2 NORMALIZED TRANSFORMER

The matrix-vector multiplication of $W_q \in \mathbb{R}^{d_{\text{model}} \times d_k}$ of the $i$-th head[1] and $h \in \mathbb{R}^{d_{\text{model}}}$ can be viewed as a dot product between the columns of $W_q$ and $h$. In the baseline Transformer, all matrices, including $W_q$ are unconstrained, leading to unbounded values in $q$. We propose to normalize $W_q$, $W_k$, $W_v$ and $W_o$ along their embedding dimension so that the computed dot products with $h$ can be interpreted as cosine similarity between unit norm vectors bounded in $[-1, 1]$. Thus, all attention matrices can be viewed as collections of normalized embedding vectors to be compared with.

While each element of $q$ and $k$ is now bounded, the norms of these two vectors can still vary. Moreover, injection of positional information by RoPE further distorts $q$ and $k$. We propose to additionally normalize $q$ and $k$, ensuring that the dot product of every query and key is under control:

$$q \leftarrow \text{Norm}(q)s_{qk} \tag{15}$$
$$k \leftarrow \text{Norm}(k)s_{qk}, \tag{16}$$

where $s_{qk} \in \mathbb{R}^{d_k}$ is a vector[2] of trainable scaling factors for the $i$-th head.

In the original Transformer, the softmax scaling factor $1/\sqrt{d_k}$ in equation 13 is introduced to account for the expected variance of $d_k$ in the dot product of non-normalized query and key vectors. In the normalized Transformer, the expected variance of the dot product between normalized query and key vectors is $1/d_k$. To restore a variance of 1, the softmax scaling factor should instead be $\sqrt{d_k}$. If the softmax scaling factor is set to 1, this is equivalent to initializing the scaling factors $s_{qk}$ at $d_k^{1/4}$.

## 2.4 MLP BLOCK

### 2.4.1 BASELINE TRANSFORMER

The input hidden state $h$ of the MLP block is first normalized using RMSNorm and then passed through two separate linear projections, producing two intermediate vectors (we omit bias terms):

$$u \leftarrow hW_u, \quad \nu \leftarrow hW_\nu \tag{17}$$

---

[1]All the following equations are defined per head but we omit $i$-th head index for the sake of readability.

[2]There is no need for separate scaling factors for $q$ and $k$ as the scaling would simply be applied element-wise when computing the dot product between $q$ and $k$.

where $\boldsymbol{W}_u, \boldsymbol{W}_\nu \in \mathbb{R}^{d_{\text{model}} \times d_{\text{MLP}}}$ are the learned weight matrices. The intermediate vectors $\boldsymbol{u}$ and $\boldsymbol{\nu}$ are combined using a gated activation function called SwiGLU defined by Shazeer (2020) as:

$$\text{SwiGLU}(\boldsymbol{u}, \boldsymbol{\nu}) \leftarrow \boldsymbol{u} \cdot \text{SiLU}(\boldsymbol{\nu}) \tag{18}$$

where $\text{SiLU}(\boldsymbol{\nu}) = \boldsymbol{\nu} \cdot \sigma(\boldsymbol{\nu})$, and $\sigma(\boldsymbol{\nu})$ is the sigmoid function. The result of the gated activation is then passed through a final linear transformation $\boldsymbol{W}_{o\text{MLP}} \in \mathbb{R}^{d_{\text{MLP}} \times d_{\text{model}}}$:

$$\boldsymbol{h}_{\text{M}} \leftarrow \text{SwiGLU}(\boldsymbol{u}, \boldsymbol{\nu}) \boldsymbol{W}_{o\text{MLP}} \tag{19}$$

### 2.4.2 NORMALIZED TRANSFORMER

We propose to normalize matrices $\boldsymbol{W}_u$ and $\boldsymbol{W}_\nu$ along the embedding dimension so that the $\boldsymbol{u}$ and $\boldsymbol{\nu}$ vectors represent the cosine similarity between $\boldsymbol{h}$ and vectors stored in $\boldsymbol{W}_u$ and $\boldsymbol{W}_\nu$, respectively. To control their impact, we introduce scaling factors $\boldsymbol{s}_u \in \mathbb{R}^{d_{\text{MLP}}}$ and $\boldsymbol{s}_\nu \in \mathbb{R}^{d_{\text{MLP}}}$:

$$\boldsymbol{u} \leftarrow \boldsymbol{u} \boldsymbol{s}_u, \tag{20}$$

$$\boldsymbol{\nu} \leftarrow \boldsymbol{\nu} \boldsymbol{s}_\nu \sqrt{d_{model}}, \tag{21}$$

where the rescaling of $\boldsymbol{\nu}$ by $\sqrt{d_{model}}$ is needed to benefit from the non-linearity of SiLU (see the Appendix A.1). The output of the MLP block is invariant to rescaling of $\boldsymbol{u}$ by a scalar.

## 2.5 EFFECTIVE LEARNING RATES IN ADAM

The core of the Adam algorithm by Kingma (2014) is as follows:

$$\begin{aligned} \boldsymbol{m} &\leftarrow \beta_1 \boldsymbol{m} + (1 - \beta_1) \boldsymbol{g} \\ \boldsymbol{v} &\leftarrow \beta_2 \boldsymbol{v} + (1 - \beta_2) \boldsymbol{g}^2 \\ \boldsymbol{\theta} &\leftarrow \boldsymbol{\theta} - \alpha \boldsymbol{m} / (\sqrt{\boldsymbol{v}} + \epsilon), \end{aligned} \tag{22}$$

where $\boldsymbol{\theta}$ is the parameter vector, $\boldsymbol{g}$ is the batch gradient, $\boldsymbol{m}$ is the momentum, $\boldsymbol{v}$ is the estimate of the per-element gradient amplitudes, $\alpha$ is the scheduled learning rate, $\epsilon$ is a small constant, and $\beta_1 < \beta_2$ are momentum factors close to 1. We cite the text of the original Adam paper using our notation: *In more common scenarios, we will have that $\frac{m}{\sqrt{v}} \approx \pm 1$ since $\left| \mathbb{E}[g] / \sqrt{\mathbb{E}[g^2]} \right| \leq 1$. The effective magnitude of the steps taken in parameter space at each timestep is approximately bounded by the stepsize setting $\alpha$.* Thus, $\alpha$ controls the effective step-size in the search space, while the ratio $\frac{m}{\sqrt{v}}$ can temporarily increase (respectively, decrease) the step-size if the current amplitude of per-parameter momentum is greater (respectively, smaller) than its estimated value over longer time horizon. Consider an example where $\theta_i = 0.01$ and the global learning rate is 0.001. If the gradient amplitude remains stable (i.e., $\frac{m_i}{\sqrt{v_i}} \approx 1$), it would take $\frac{0.02 - 0.01}{0.001} = 10$ steps to double $\theta_i$. However, if $\theta_i = 1.0$, it would take $\frac{2.0 - 1.0}{0.001} = 1000$ steps to double. Even if the gradient's amplitude is larger in the second case, the number of steps would only decrease if $m_i > \sqrt{v_i}$.

In nGPT, for any trainable vector of scaling parameters such as $\boldsymbol{s}_a$, we use two scalars $\boldsymbol{s}_{a,init}$ and $\boldsymbol{s}_{a,scale}$. When initializing $\boldsymbol{s}_a$ as a trainable parameter, its initial value is set to $\boldsymbol{s}_{a,scale}$. However, during the forward pass we restore its actual value by multiplying $\boldsymbol{s}_{a,init}/\boldsymbol{s}_{a,scale}$. This allows us to control the effective learning rate for $\boldsymbol{s}_a$ by adjusting $\boldsymbol{s}_{a,scale}$, while keeping the global learning rate unchanged. For example, setting $\boldsymbol{s}_{a,init} = 1$ and $\boldsymbol{s}_{a,scale} = 1/\sqrt{d_{model}}$ ensures that this parameter is updated with the same effective learning rate as other normalized parameters in the network.

## 2.6 SUMMARY OF MODIFICATIONS

The recipe to convert the baseline Transformer into the normalized Transformer is as follows:

1. Remove all normalization layers such as RMSNorm or LayerNorm.
2. After each training step (and, optionally, during the forward pass), normalize matrices $\boldsymbol{E}_{\text{input}}, \boldsymbol{E}_{\text{output}}, \boldsymbol{W}_q, \boldsymbol{W}_k, \boldsymbol{W}_v, \boldsymbol{W}_o, \boldsymbol{W}_u, \boldsymbol{W}_\nu$ and $\boldsymbol{W}_{o\text{MLP}}$ along their embedding dimension.
3. Replace the update equations 4 and 5 by equations 10 and 11, where $\boldsymbol{\alpha}_{\text{A}}$ (and also $\boldsymbol{\alpha}_{\text{M}}$) is treated with $\boldsymbol{\alpha}_{\text{A},init} = 0.05$ (in order of $1/n_{layers}$) and $\boldsymbol{\alpha}_{\text{A},scale} = 1/\sqrt{d_{model}}$.

4. Change the softmax scaling factor in attention from $1/\sqrt{d_k}$ to $\sqrt{d_k}$. Implement the rescaling and normalization (normalization here is optional) of $\boldsymbol{q}$ and $\boldsymbol{k}$ as in equations 15 and 16, where $\boldsymbol{s}_{qk}$ is treated with $\boldsymbol{s}_{qk,init} = 1$ and $\boldsymbol{s}_{qk,scale} = 1/\sqrt{d_{model}}$.

5. Implement the rescaling of the intermediate state of the MLP block using equations 20 and 21, where $\boldsymbol{s}_u$ (and also $\boldsymbol{s}_\nu$) is treated with $\boldsymbol{s}_{u,init} = 1$ and $\boldsymbol{s}_{u,scale} = 1$

6. Implement the rescaling of logits using equation 3, where $\boldsymbol{s}_z$ is treated with $\boldsymbol{s}_{z,init} = 1$ and $\boldsymbol{s}_{z,scale} = 1/\sqrt{d_{model}}$.

7. Remove weight decay and learning rate warmup.

# 3 EXPERIMENTS

We train both the baseline Transformer (GPT) and the normalized Transformer (nGPT) on the Open-WebText dataset (Gokaslan & Cohen, 2019) and evaluate them on a set of standard downstream tasks. We experiment with models containing 0.5B and 1B parameters, including the embeddings. For both GPT and nGPT, we report results using the best initial learning rate settings (see Appendix A.7). A detailed description of the setup and hyperparameters is in Appendix A.6.

## 3.1 ACCELERATION OF TRAINING

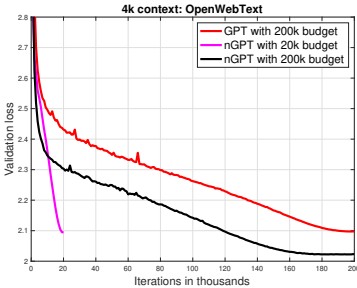

Figure 1: Validation loss during training of 1B GPT and nGPT with 4k context length.

Figure 1 presents the validation loss during the training of GPT and nGPT models with 1 billion parameters and a sample length of 4k tokens. After 20k iterations, nGPT achieves the same validation loss that GPT reaches only after 200k iterations (approximately 400 billion tokens), demonstrating a 10x speedup in terms of iterations and tokens used.[3]

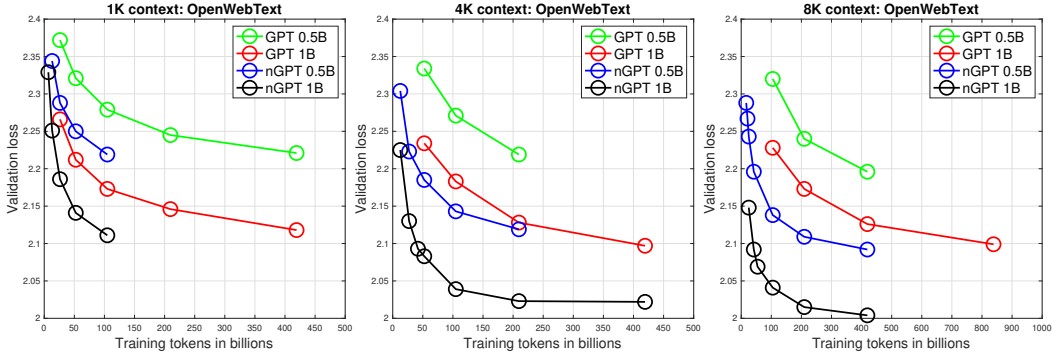

Figure 2: Final validation loss (**y-axis**) for training runs with different computation budgets in tokens (**x-axis**). The training of 0.5B and 1B nGPT models is about 4x, 10x and 20x faster (in terms of tokens) on 1k, 4k and 8k context lengths, respectively.

---

[3]While the time per step of nGPT is higher (80% - for 4k, and 60% - for 8k context respectively), it can be reduced after code optimization. Also the overhead is less significant for larger networks (see Appendix A.5).

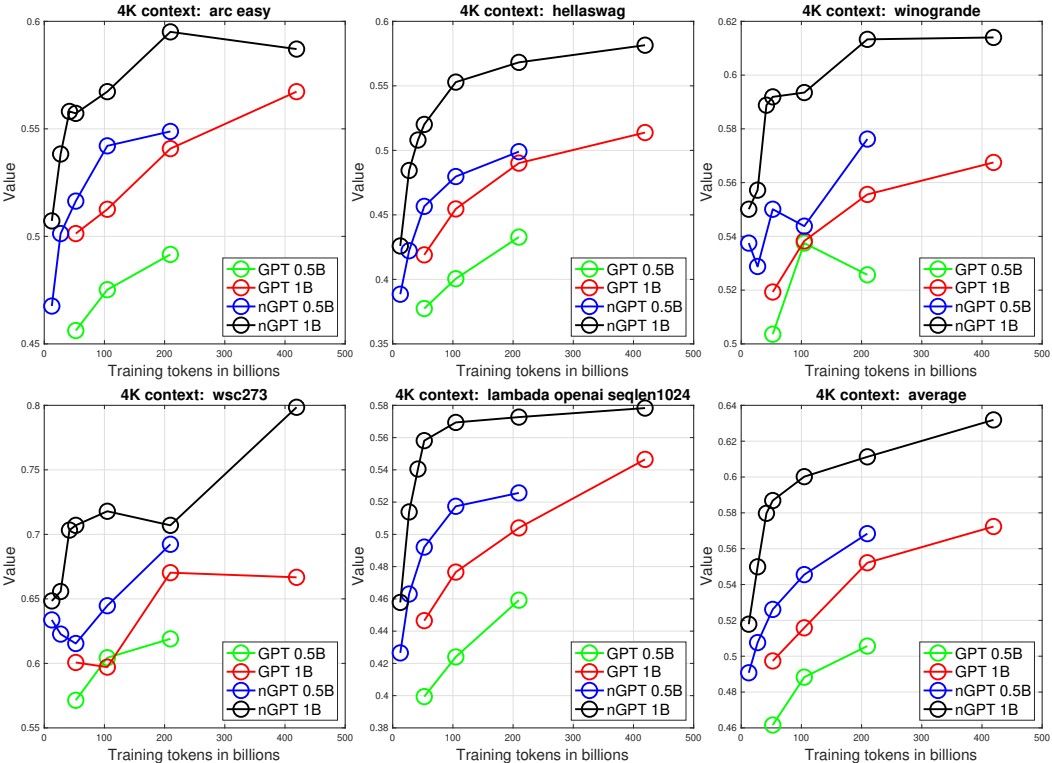

Figure 3: Models trained with 4k context length. Final performance (**y-axis**) on a set of downstream tasks and their average value (**Bottom-Right**) for different computation budgets in tokens (**x-axis**).

Figure 2 illustrates how the performance gap between nGPT and GPT scales across three axes: total token budget, context length, and network size. Training the 0.5B and 1B nGPT models is approximately 4x, 10x, and 20x faster at context lengths of 1k, 4k, and 8k tokens, respectively.

Figure 3 shows a similar pattern across downstream tasks, confirming that the acceleration is not only reflected in perplexity but also in task performance. Figures 8 and 10 in the Appendix provide results for 1k and 8k context lengths. We observe some saturation for the longest runs of nGPT, suggesting that the model capacity is nearly reached for this number of trainable model parameters.

## 3.2   INSPECTION OF NETWORK PARAMETERS

Figure 4 shows that, while nGPT maintains a fixed norm for embeddings (by design), GPT exhibits significant variation. The distribution of eigenvalues, computed from the covariance matrix of embeddings and normalized by their median, reveals that GPT's input embeddings have a higher condition number, especially in the 1B model. The distribution of pairwise dot products between embeddings indicates that even in nGPT, embeddings are not uniformly distributed across the hypersphere (where the dot product would approach 0), but instead form clusters—possibly reflecting natural patterns in language data. Dot products in GPT tend to have higher values due to its embeddings forming a hyper-ellipsoid, as suggested by the spread of vector norms. The ill-conditioned nature of GPT's input embeddings could lead to computational issues involving these embeddings.

Figure 5 shows the median condition numbers (across heads) for attention and MLP matrices at different layer depths—24 layers for the 0.5B model and 36 layers for the 1B model. GPT models exhibit significantly higher condition numbers in their attention matrices compared to nGPT. A closer inspection of these matrices (the 3rd and 4th layers are in Figure 12 and Figure 13 of Appendix) suggests that they degenerate into lower-rank matrices, potentially reducing the learning capacity of these blocks. One could argue that the elevated condition numbers are influenced by the norms of the vectors in these matrices. Our post-training normalization of these matrices is depicted

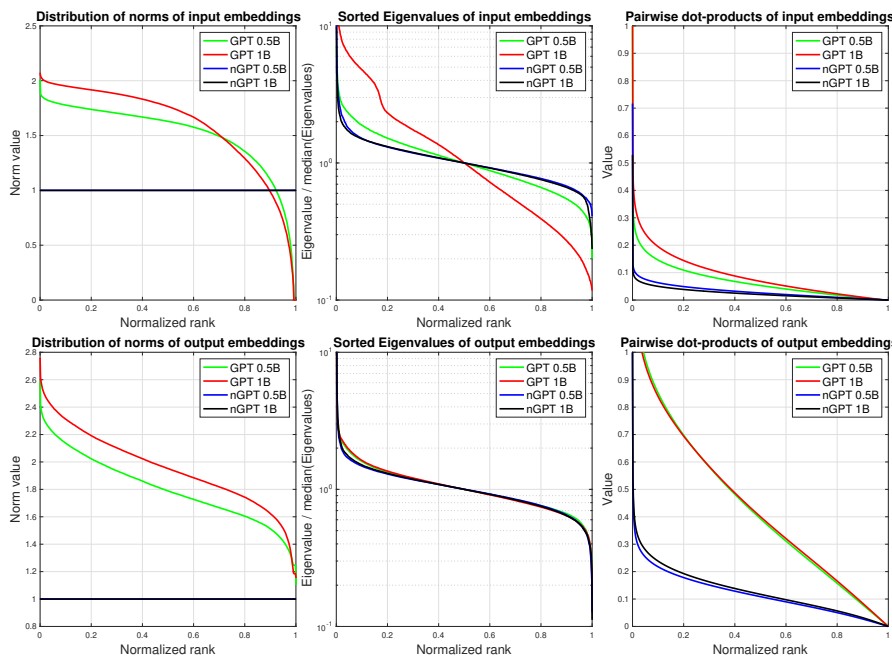

Figure 4: **Left**: Distribution of norms of vectors from input (**Top line**) and output (**Bottom line**) embedding matrices. **Middle**: Distribution of eigenvalues divided by its median value. **Right**: Pairwise distribution of dot products between embeddings. Models are trained for 100k iterations.

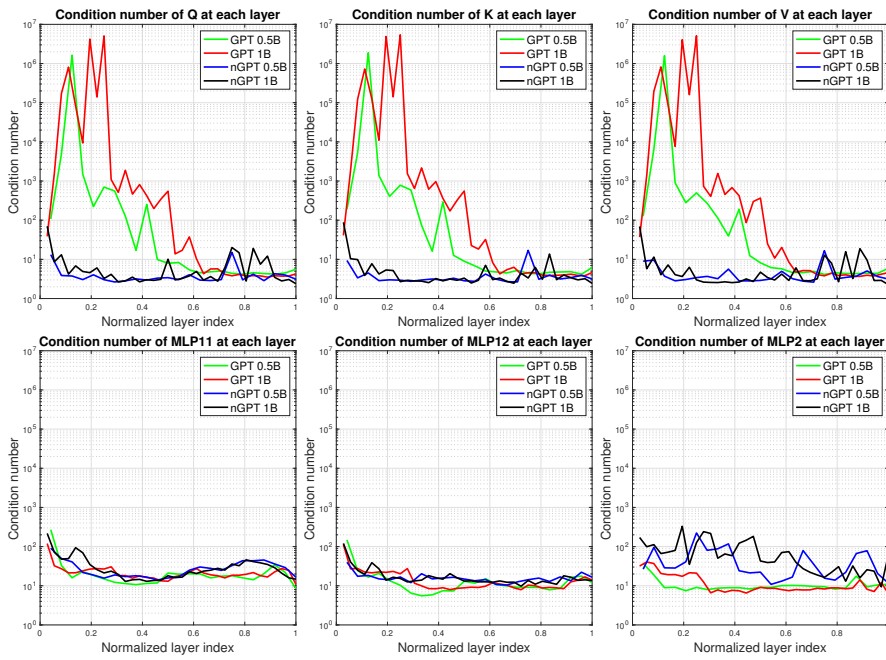

Figure 5: Median condition numbers for attention and MLP matrices at different layer depth (24 and 36 layers for 0.5B and 1B models, respectively). Models are trained for 100k iterations.

by the dotted lines in Figure 11 of the Appendix. While the adjusted condition numbers are reduced, they remain higher than those for nGPT, indicating potential rank deficiency. The need for such normalization highlights one of the issues that nGPT is specifically designed to address.

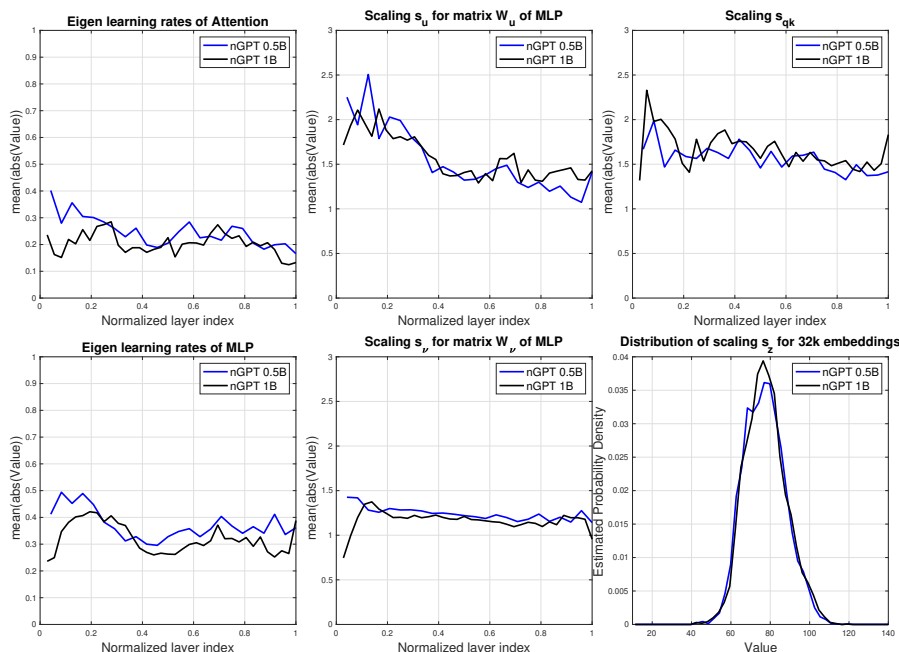

Figure 6: (**Left**): Eigen learning rates of Attention and MLP blocks. (**Middle**): Scaling factors applied to the intermediate states of MLP. (**Right**): Scaling factors applied before the QK dot product; distribution of per-vector scalings applied to logits. Models are trained for 100k iterations.

An important contribution of this work is the decoupling of predictions made by the Attention and MLP blocks from their impact on the hidden state $h$. These contributions are controlled by the eigen learning rates $\alpha_A$ and $\alpha_M$. Their interpretation is straightforward: if $\alpha_{A,i}$ for an embedding dimension $i \in \mathbb{R}^{d_{model}}$ is 0.2, then the update follows $h_i \leftarrow (1-0.2)h_i + 0.2h_{A,i}$. Thus, they directly quantify the contribution of $h_{A,i}$ into $h$. Figure 6 shows the average absolute values of $h_A$ and $h_M$ at each layer. Notably, the network learns to take only modest steps (20%-30%) in the direction suggested by $h_A$ and $h_M$. The average magnitude of $\alpha_A$ decreases from 0.25 in the 0.5B network (24 layers) to 0.20 in the 1B network (36 layers). Meanwhile, $\alpha_M$ decreases from 0.37 to 0.32. If the parameter count per block is linked to its maximum predictive capacity (in our setup, MLP blocks have more parameters than Attention blocks), then the greater eigen learning values observed for MLPs could potentially be attributed to the higher quality predictions made by these blocks.

The scaling factors $s_u$, $s_\nu$ and $s_{qk}$ remain relatively stable across layers. The value of $s_\nu$ can be interpreted as a measure of the non-linearity of the SiLU function, which behaves like ReLU for large $s_\nu$ and approximates a linear unit for values near 0 (see also Appendix A.1). The distribution of $s_z$ is primarily characterized by its mean, which influences the temperature of the softmax during cross-entropy calculations. The introduced scaling factors $s_{qk}, s_u, s_\nu$ and $s_z$ seem to compensate for the removal of magnitude information when normalizing matrices and embeddings.

## 3.3 ABLATION STUDIES

Appendix A.9 summarizes numerous ablation experiments. An important finding is that having fixed (non-learnable) values for $s_{qk}, s_u, s_\nu$ and a single global learnable value for $s_z$ leads to only a slight degradation in accuracy. Therefore, our presented general case can be simplified and become easier to interpret. Appendix A.8 demonstrates that nGPT can handle longer contexts without requiring any modifications to RoPE.

## 4 RELATED WORK

Wang & Isola (2020) provides a comprehensive overview of the arguments for representation learning on the hypersphere. Spherical representations are associated with more stable training in the latent space of variational autoencoders (Xu & Durrett, 2018) and in embeddings used for face verification (Wang et al., 2017). Notably, when embeddings are well clustered, they tend to be linearly separable from the rest of the embedding space (Wang & Isola, 2020). Mettes et al. (2019) demonstrated that classification and regression can be unified by placing prototype embeddings uniformly on a hypersphere, allowing for separation with large margins a priori. Wang & Isola (2020) found a strong empirical correlation between downstream task performance and both the alignment (closeness) and uniformity of embeddings on the hypersphere.

Since all embeddings in nGPT lie on the hypersphere, any update that causes the hidden state $h$ to deviate from the manifold is followed by a normalization step. This normalization can be interpreted as a retraction in the context of Riemannian optimization. One might attempt to approximate nGPT's update in GPT by applying RMSNorm both at the beginning and end of the block (Xiong et al., 2020). However, this approach does not guarantee a fixed norm for the hidden state, nor does it ensure that the recombination approximates SLERP or LERP.

The normalization in equations 15 and 16 closely resembles the QK normalization by Henry et al. (2020). In nGPT, this process can be viewed as restoring $q$ and $k$ of the $i$-th head to a $(d_{\text{model}}/n_{\text{heads}})$-dimensional hypersphere after the projection of $h$ by $W_q$ and $W_k$, respectively. Since $h$ and the embedding dimensions of $W_q$ and $W_k$ are already normalized, the norms of $q$ and $k$ are also comparable, making their normalization potentially unnecessary. We investigated the effect of omitting this normalization in our ablation studies (see Appendix A.9). The results indicate only a minor performance degradation with potential computational savings of about 12%. However, the normalization helps to maintain performance when extrapolating the context length (see Appendix A.8).

## 5 DISCUSSION AND CONCLUSION

This work builds on numerous key findings and observations made in the field which directly (Liu et al., 2017; Wang et al., 2017; Liu et al., 2018; Xu & Durrett, 2018; Wang & Isola, 2020; Liu et al., 2021; Karras et al., 2024) and indirectly (Salimans & Kingma, 2016; Franke et al., 2023; Kodryan et al., 2022; Kosson et al., 2023) support representation learning on the hypersphere. One of our main contributions is the normalization of the embedding dimensions across all of the transformer's matrices, ensuring they reside on the same hypersphere. Crucially, we observed that such normalization alone would constrain the inputs of non-linear units, and, thus, the scaling factors for these units should be introduced.

In line with recent studies suggesting that transformers implicitly perform gradient descent as meta-optimizers (Von Oswald et al., 2023; Dai et al., 2022), we explicitly demonstrate how this process occurs in the normalized Transformer: i) the transformation blocks provide gradient information, ii) this information is multiplied by eigen learning rates to adjust the hidden state, and iii) the commonly used normalization can be interpreted as a retraction step in Riemannian optimization, projecting the point back onto the hypersphere. We believe we are the first to decouple the eigen learning rates from the rest of the network, recognizing them as trainable parameters that can be interpreted as the diagonal elements of a variable-metric matrix. In other words, the normalized Transformer functions as a variable-metric optimizer, searching for output solutions using data-driven gradient information estimated in its attention and MLP blocks.

The spherical representation provides valuable insights into the internals of nGPT, enabling the collection and analysis of statistics about its normalized components. Most importantly, it allows for the application of mathematical techniques specifically designed for dealing with hyperspheres. We believe that the reported acceleration, by a factor from 4 to 20, is only the first step towards uncovering new algorithms and architectures that could emerge from nGPT. Future work should explore scaling nGPT to larger network sizes, real-world datasets, and a broader range of tasks. For instance, the extension of nGPT to encoder-decoder and hybrid architectures (Dao & Gu, 2024; De et al., 2024) is straightforward.

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

## A   APPENDIX

### A.1   RESCALING IN THE MLP BLOCK OF THE NORMALIZED TRANSFORMER

When computing SwiGLU using equation 18, each element $x$ of the vector $\boldsymbol{v}$ is an input to SiLU:

$$\mathrm{SiLU}(x) = x \cdot \sigma(x) = x \cdot \frac{1}{1 + e^{-x}}, \tag{23}$$

where $\sigma(x)$ is sigmoid. For $x$ with large magnitude, $\mathrm{SiLU}(x)$ approximates $\mathrm{ReLU}(x)$: when $x \to -\infty$, $\mathrm{SiLU}(x) \to 0$, and when $x \to \infty$, $\mathrm{SiLU}(x) \approx x$. The minimum of $\mathrm{SiLU}(x_{min}) \approx -0.278$ is located at $x_{min} \approx -1.278$. While the elements of $\boldsymbol{v}$ represent dot products of $d_{model}$-dimensional vectors and are bounded in $[-1, 1]$, their expected absolute value (when they are random) is $\mathbb{E}[|\cos(\theta)|] = \frac{2}{\pi} \cdot \frac{1}{\sqrt{d_{model}}} \approx \frac{0.7979}{\sqrt{d_{model}}}$. Thus, we should rescale $x$ by $\sqrt{d_{model}}$, otherwise, for very small $x$, we end up with $\mathrm{SiLU}(x) \approx x/2$. An alternative view is to note that since the variance of each of the normalized vectors ($\boldsymbol{h}$ and a vector from $\boldsymbol{W}_v$) is $1/d_{model}$, the variance of 1 (suitable for the sigmoid part of SiLU) can be restored by rescaling of $x$ by $\sqrt{d_{model}}$. Based on these two views, we rescale $\boldsymbol{v}$ by $\sqrt{d_{model}}$ to benefit from the non-linearity of SiLU.

### A.2   EIGEN LEARNING RATES

In the main text of the paper, we defined eigen learning rates as positive ($\boldsymbol{\alpha} \leftarrow |\boldsymbol{\alpha}|$ is used during the forward pass). However, when they are not constrained to be positive, we obtain experimental results which are the same (up to numerical difference). This surprising observation can be explained as follows. Both the attention and MLP blocks have transformation matrices at their outputs. When the search is unconstrained, it is sufficient for Adam to flip (select) the sign of the $i$-th row of the output transformation matrix $\boldsymbol{W}_o$ to change the sign of the corresponding $i$-th coordinate in $\boldsymbol{h}_{\mathrm{A}}$. Thus, the transformation calculated as $\boldsymbol{\alpha}_{\mathrm{A}} \boldsymbol{W}_o$ is the same as $\boldsymbol{\alpha}'_{\mathrm{A}} \boldsymbol{W}'_o$, where $\boldsymbol{\alpha}'_{\mathrm{A},i} = -\boldsymbol{\alpha}_{\mathrm{A},i}$ and $\boldsymbol{W}'_{o,(i,:)} = -\boldsymbol{W}_{o,(i,:)}$. In other words, when unconstrained, we can arrive at exactly the same transformation by flipping the signs in both $\boldsymbol{\alpha}$ and $\boldsymbol{W}_o$, which cancel each other. For simplicity and clearer interpretation, we suggest constraining $\boldsymbol{\alpha}$ to be positive in the main paper.

There is, however, another interesting scenario when eigen learning rate could become negative. In quasi-Newton methods, $\boldsymbol{B}$ approximates the inverse Hessian matrix $\boldsymbol{H}^{-1}$ whose diagonal elements are positive values if the function is locally convex which is the assumption of the Newton method ($\boldsymbol{H}$ needs to be positive definite). However, the diagonal of $\boldsymbol{H}^{-1}$ can have negative values if the objective function is non-convex and has saddle points. While quasi-Newton methods like BFGS aim to ensure (e.g., via regularization) that $\boldsymbol{B}$ is positive-definite even on non-convex problem, some Riemannian optimization methods can exploit negative curvature of $\boldsymbol{H}^{-1}$ (Agarwal et al., 2021). When the diagonal of $\boldsymbol{B}$ is unconstrained, we perform a variable-metric step, acknowledging that the function may be non-convex locally. We did not mention this in the main paper because, as noted earlier, the results with both constrained and unconstrained $\boldsymbol{\alpha}$ are essentially the same.

During the review process of this paper, we were made aware of the work of Bachlechner et al. (2020) which proposes ReZero (residual with zero initialization). In the context of Transformers, ReZero is implemented as:

$$\boldsymbol{h} \leftarrow \boldsymbol{h} + \alpha \boldsymbol{h}_{\mathrm{T}} \tag{24}$$

where the hidden state $\boldsymbol{h}$ at each layer is updated with $\alpha$-rescaled output $\boldsymbol{h}_{\mathrm{T}}$ of the transformation block. The trainable layer-specific scalar $\alpha$ is the same for both the MLP and Attention blocks, with its initial value suggested to be set to zero. The authors also proposed to remove LayerNorm and suggested that BatchNorm is a suitable alternative. We note that the introduction of the scaling factor $\alpha$ does not affect the architecture of the model because the output projection matrices of both MLP and Attention blocks already contain that same scaling factor as a multiplicative part of their parameters (e.g., a simultaneous rescaling of $\alpha$ by $k$ and the output projection matrices by $1/k$ results in a network with identical behavior). However, the use of a single scaling factor can help

to more quickly adjust the impact of each transformation block. Indeed, the authors argue that by initializing $\alpha$ to zero, one can achieve a form of curriculum learning in which individual layers of the network can be gradually activated during optimization.

The update of ReZero differs from the update of nGPT which can be written as:

$$\boldsymbol{h} \leftarrow \text{Norm}(\, \boldsymbol{h} + \boldsymbol{\alpha}_\text{T}(\text{Norm}(\boldsymbol{h}_\text{T}) - \boldsymbol{h})\,), \tag{25}$$

where the original output of the transformation block $\boldsymbol{h}_\text{T}$ is normalized before being used to compute an update direction, weighted by the eigen learning rate of the block $\boldsymbol{\alpha}_\text{T}$. In contrast to ReZero, the update of nGPT is invariant to any rescaling of the output projection matrix, and, thus, the eigen learning rate can be viewed as a measure of the contribution of the block. This is not the case for ReZero, where the scale of $\alpha$ cannot be viewed as a measure of the impact of the block without taking into account the scaling of the block's matrices (see, e.g., our previous example with rescaling by $k$). In contrast to ReZero, where $\alpha$ rescales $\boldsymbol{h}_\text{T}$, the update of nGPT rescales the direction towards $\boldsymbol{h}_\text{T}$. Finally, the update of nGPT holds for the spherical representation where the updated hidden state is retracted back to the hypersphere.

Another work that we were made aware of during the review process is NormFormer (Shleifer et al., 2021) which updates the hidden state as follows:

$$\boldsymbol{h} \leftarrow \boldsymbol{h} + \text{LN}(\, \boldsymbol{h}_\text{A}\,) \tag{26}$$
$$\boldsymbol{h} \leftarrow \boldsymbol{\alpha}\boldsymbol{h} + \text{LN}(\, \sigma(\text{LN}(\boldsymbol{h}))\boldsymbol{W}_1)\boldsymbol{W}_2, \tag{27}$$

where $\boldsymbol{h}_\text{A}$ is the output of the Attention block with LayerNorm normalization of inputs, $\boldsymbol{W}_1$ and $\boldsymbol{W}_2$ matrices are components of the MLP block with GELU activation function, and, $\boldsymbol{\alpha}$ are trainable scaling factors defined parameter-wise. If we omit the variance part of LayerNorm in Equation 26, then $\text{LN}(\, \boldsymbol{h}_\text{A}\,)$ can be reshaped to $\boldsymbol{\alpha}\text{Norm}(\, \boldsymbol{h}_\text{A}\,)$ and one could wonder if $\boldsymbol{\alpha}$ can be viewed as eigen learning rates. This is not the case because, similarly to ReZero, i) the norm of $\boldsymbol{h}$ is not constrained, and ii) the scaling is applied to the value itself and not to the direction. In Equation 27, not only the norm of $\boldsymbol{h}$ is not constrained, but the output of the MLP is also not normalized. Instead, LayerNorm normalizes the rescaling factors for the vectors stored in $\boldsymbol{W}_2$. While the name NormFormer may suggest some similarities with nGPT, the approach does not normalize network parameters in any way and does not normalize embeddings during the update over layers (i.e., the typically growing norm of $\boldsymbol{h}$ affects its recombination with the outputs of the transformation blocks).

### A.3    Supplementary Notes to Introduction

The original Transformer architecture is a solution in some design space which can be measured with respect to a set of metrics. Many known aspects of multi-criteria decision-making and multi-objective optimization are applied (Deb et al., 2016). Various Transformer variants exist because depending on the selection of metrics (e.g., performance on particular tasks) and constraints (e.g., memory, compute, data), the decision-maker prefers and selects (e.g., based on non-dominance or aggregation of objectives) different design solutions. The first part of the Introduction section describes various architectural and optimization findings/solutions of the past. Then, it is proposed to "unify various findings and observations made in the field under a new perspective of the normalized Transformer." This unification can be viewed as a recombination procedure in the design space of architectures and optimization approaches. The Introduction section concludes with a description of the key aspects of the proposed solution. The experimental results of the paper demonstrate some advantages of this solution w.r.t. the baseline GPT, e.g., improved condition number of attention matrices, faster convergence.

While the proposed solution can be viewed as a recombination of existing techniques, this is not how it was historically designed. The precursor of nGPT is described in Loshchilov (2023), where a modification of AdamW was studied in the training of Transformer models. That work demonstrated that instead of performing weight decay, the norm of all parameters of the network can be controlled and scheduled over the course of training. While this approach enabled acceleration in training, it was noted that the attention matrices are ill-conditioned and the normalization of the whole network or even individual matrices does not resolve it. In contrast, experiments with normalization of

individual vectors within matrices demonstrated improvements of the condition numbers. Then, by combining this finding with the understanding that LayerNorm and RMSNorm normalize the hidden state to a hypersphere (when all scaling factors are equal), it was hypothesized that the spherical representation is a viable option. nGPT was designed from first principles, focusing on ensuring that the Transformer architecture respects the spherical representation for all its vector components. SLERP was identified as the proper way to recombine the hidden state and the output of the transformation blocks when working on the hypersphere and using scalar eigen learning rates. Scaling factors, such as those used for logits, were introduced to relax the constraints imposed by the spherical formulation.

During the review process of this paper, we were made aware of a set of closely related works. Karras et al. (2024) introduces a set of normalization techniques (very similar to ours) which greatly improve the training of Diffusion Models. Liu et al. (2021) proposed an optimization method for neural networks under hard hyperspherical weight constraints. Large et al. (2025) employs a residual block structure which is related to the one of nGPT but without learnable parameters (eigen-learning rates). By connecting transformers to modern Hopfield models, Wu et al. (2024) show that well-separated memory patterns (learned representations) enhance memorization capability (expressiveness), naturally achieved by mapping learning onto the hypersphere. The authors conclude that representation learning on the unit hypersphere significantly improves expressiveness and memorization. Hu et al. (2024) claim that mapping to the hypersphere makes representation learning provably optimal for any dataset.

### A.4 RIEMANNIAN OPTIMIZATION

If $h - h_A$ is viewed as the gradient $g$ in the Euclidean space, then, aligning with the requirements of Riemannian optimization, the projection of $g$ onto the tangent space of the hypersphere is

$$g_{\text{proj}} \leftarrow h(h^T h_A) - h_A \tag{28}$$

The projection is equivalent to $g$ when the dot product $h^T h_A$ is 1. Depending on the alignment between $h$ and $h_A$, the projected gradient varies between $h - h_A$ (when the vectors are aligned) and $-h_A$ (when the vectors are orthogonal). The Riemannian variable-metric update then reads as:

$$h \leftarrow \text{Norm}(\, h - B_A(h(h^T h_A) - h_A)\,) \tag{29}$$

$$h \leftarrow \text{Norm}(\, h - B_M(h(h^T h_M) - h_M)\,) \tag{30}$$

The normalization by Norm can be viewed as the retraction step in Riemannian optimization, mapping the updated solution back to the manifold.

Our experimental results suggest that the impact of $h^T h_M$ is negligible. Therefore, all experiments in the paper are based on equations 10 and 11.

### A.5 TIME COST PER STEP

The time cost per step for nGPT is approximately 80% higher with 4k context length, and 60% higher with 8k context length. This overhead is not only due to nGPT having 6 normalization steps (2 of them are applied for $q$ and $k$) per layer instead of 2, but also because nGPT's normalizations are not yet fully optimized, unlike GPT, where normalization layers are fused with other operations. Training on larger networks is expected to further reduce this performance gap, as the number of layers (and thus the number of normalizations) increases only modestly with the number of network parameters. Appendix A.9 shows that we can remove the normalization of $q$ and $k$ with a minor negative impact on results.

### A.6 EXPERIMENTAL SETUP

In all experiments, we use OpenWebText (Gokaslan & Cohen, 2019) dataset which, according to experiments of Karpathy (2023), represents a good approximation of the OpenAI's internal dataset

Table 2: Model Parameters for GPT and nGPT

| Model Parameter | 0.5B Models | 1.0B Models |
|---|---|---|
| Number of Layers ($n_{\text{layers}}$) | 24 | 36 |
| Model Dimension ($d_{\text{model}}$) | 1024 | 1280 |
| Number of Attention Heads ($n_{\text{heads}}$) | 16 | 20 |
| Key Dimension ($d_k$) | $d_{\text{model}}/n_{\text{heads}}$ | $d_{\text{model}}/n_{\text{heads}}$ |
| MLP Dimension ($d_{\text{MLP}}$) | $4d_{\text{model}}$ | $4d_{\text{model}}$ |
| Parameters in GPT | 468.2M | 1025.7M |
| Parameters in nGPT | 468.4M | 1026.1M |

Table 3: Optimization Parameters for GPT and nGPT

| Optimization Parameter | GPT | nGPT |
|---|---|---|
| Optimizer | AdamW | Adam (AdamW with weight decay 0.0) |
| Weight Decay | 0.1 | 0.0 |
| Number of Warmup Steps | 2000 | 0 |
| Learning Rate Schedule | Cosine Annealing | Cosine Annealing |
| Initial Learning Rate | problem-specific | problem-specific |
| Final Learning Rate | 0 | 0 |

used to train GPT-2 models. We are well aware that this dataset is not of the highest quality. However, we believe that it is suitable for academic research, and, should improve the comparability of our findings with other research papers.

We trained our models using 64 A100 GPUs distributed across 8 nodes (8 GPUs per node). Global batch size is 512. We use the LLaMA-2 tokenizer with 32k tokens. We use the same setup for the 0.5B and 1.0B parameter models. All parameters of matrices are stored in bfloat16.

All matrix parameters are initialized by sampling from a zero-mean normal distribution with a standard deviation of 0.02 for GPT and $1/\sqrt{d_{\text{model}}}$ for nGPT. The standard deviation for the output matrices was scaled by a factor of $\sqrt{2 \times n_{\text{layer}}}$, as suggested by Radford et al. (2018). The initialization of matrix parameters is not important for nGPT because they are normalized afterwards. The base of RoPE is 10000. The initialization of the additional parameters introduced in nGPT is described in Section 2.6.

All experiments described in this paper were performed using an internal library based on Megatron-LM (Shoeybi et al., 2019). In order to illustrate how nGPT works, we re-implemented nGPT using nanoGPT (Karpathy, 2023) and published our re-implementation at https://github.com/NVIDIA/ngpt. It should be noted that the latter re-implementation only qualitatively replicates our internal experiments and should not be used in production. However, it should be great to quickly get familiar with nGPT as its core functionality represents only a few dozens of lines of codes. It is very important to note that when implementing nGPT in training libraries, one should make sure that not only instantiated model parameters are normalized but also the ones which are used by the optimizer. Missing the latter is a common bug that should be avoided.

## A.7    Selection of the initial learning

The initial learning rate is the only hyperparameter we tune for both GPT and nGPT. Figure 7 demonstrates our trials to select the most suitable initial learning rate for GPT and nGPT. Our first experiments started with the 0.5B model and 1k context length. After observing the general trend of these curves, we reduced the number of experiments and followed the trends to minimize the total compute used. Apart from estimating the optimal settings for the initial learning rate, our general observation is that the optimal learning rates for GPT and nGPT tend to be similar for the same values of the validation loss. Longer runs are usually associated with the possibility of achieving lower values of the validation loss, and this typically requires lower initial learning rates.

One artifact we observed is the increasing sensitivity of the 1B nGPT model on 8k context length. We found that the smoothness of the hyperparameter curves can be restored (see the dotted lines with squares) by increasing $\alpha_{\text{A,init}}$ and $\alpha_{\text{M,init}}$ from their default value of $1/\sqrt{d_{model}}$ to 0.1. This change decreases the effective learning rate of Adam on these variables by a factor of $(1/\sqrt{d_{model}})/0.1 \approx 3$. As a result, the eigen learning rates are learned by Adam at a slower rate.

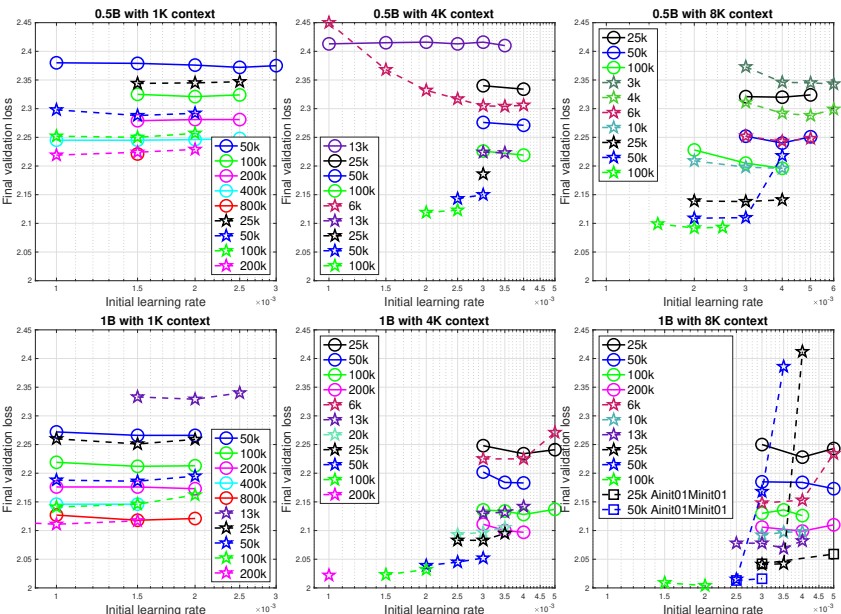

Figure 7: Final validation loss values for different initial learning rates for the 0.5B models (**Top**) and 1B models (**Bottom**). GPT is denoted by solid lines with circles, while nGPT is represented by the dotted lines with stars. A specific case with a different setup, denoted by the dotted lines with squares (**Bottom Right**), is discussed in the text.

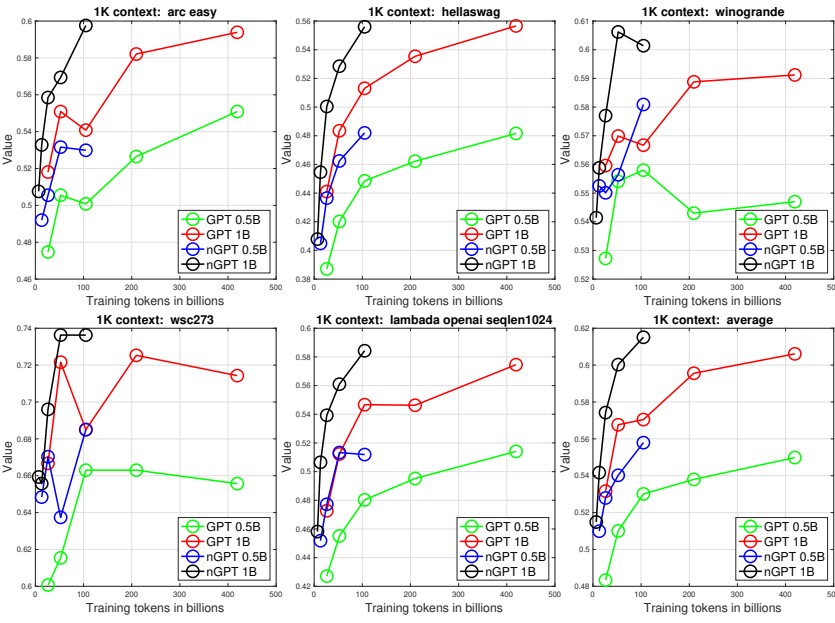

Figure 8: Models trained with 1k context length. Final performance (**y-axis**) on a set of downstream tasks and their average value (**Bottom-Right**) for different computation budgets in tokens (**x-axis**).

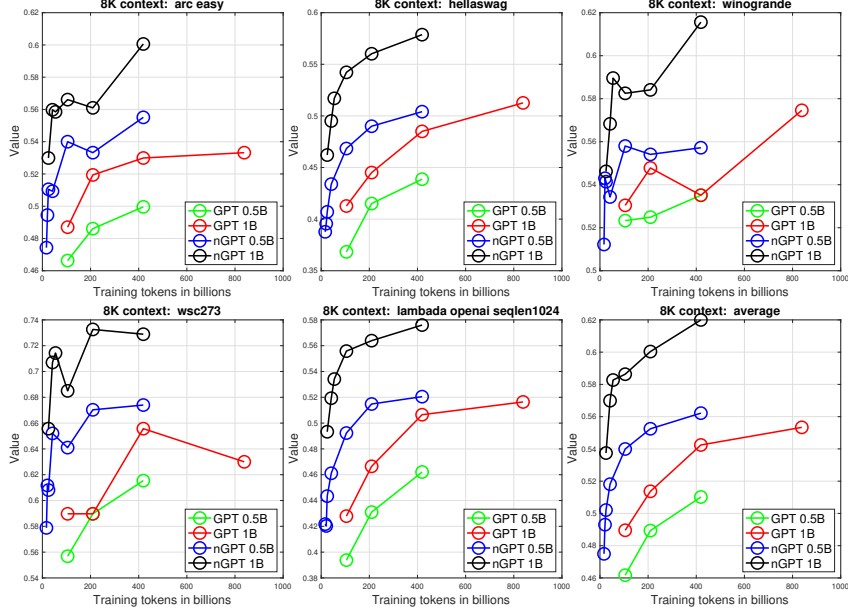

Figure 9: Models trained with 8k context length. Final performance (**y-axis**) on a set of downstream tasks and their average value (**Bottom-Right**) for different computation budgets in tokens (**x-axis**).

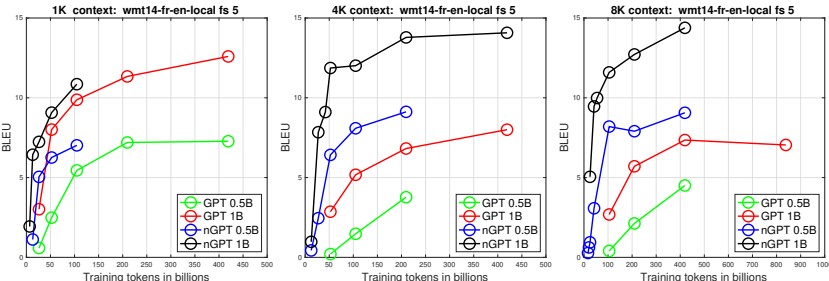

Figure 10: Performance on WMT14-FR-EN task which evaluates the ability (measured by the BLEU score of Papineni et al. (2002)) to translate a French sentence to English given five (French, English) example pairs, following the methodology of Radford et al. (2018) where it was used to benchmark GPT-2.

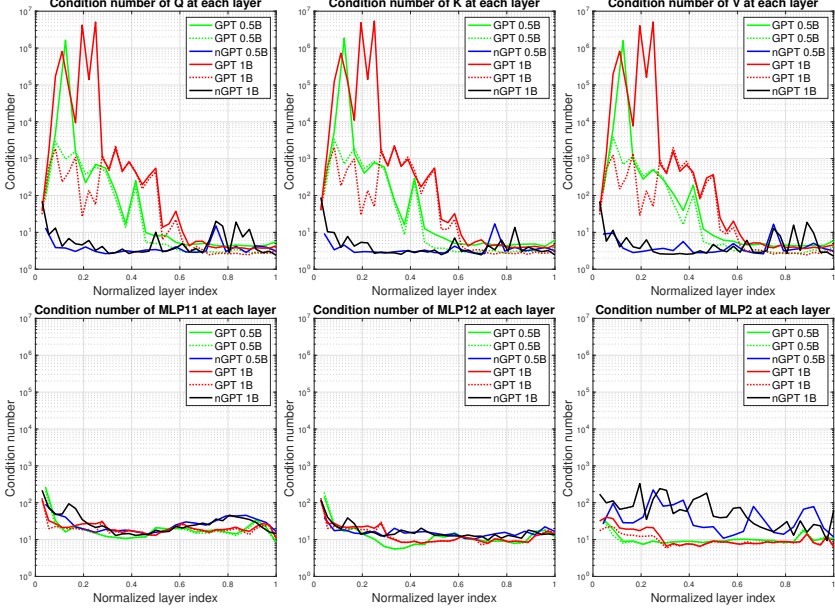

Figure 11: Median condition numbers measured for attention and MLP matrices at different layer depth (24 and 36 layers for 0.5B and 1B networks, respectively). The dotted lines are for the case when GPT's matrices are renormalized after training. Models are trained for 100k iterations.

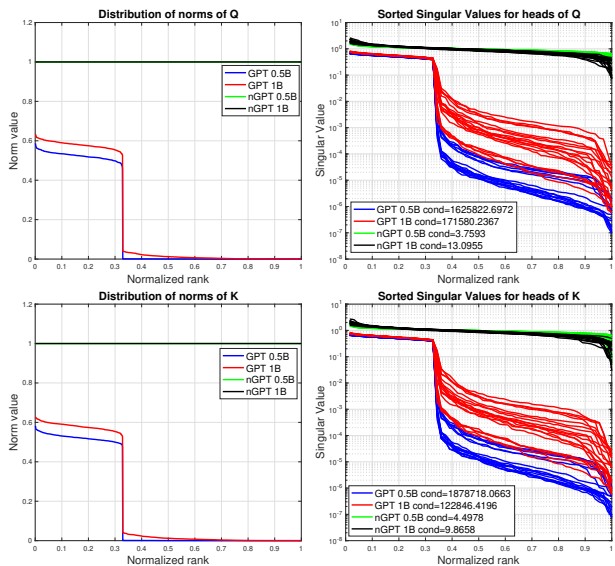

Figure 12: **Left**: Distribution of the norms of vectors forming the complete attention matrices (not per head). **Right**: Distribution of singular values for all heads. The results are shown for the 3rd layer.

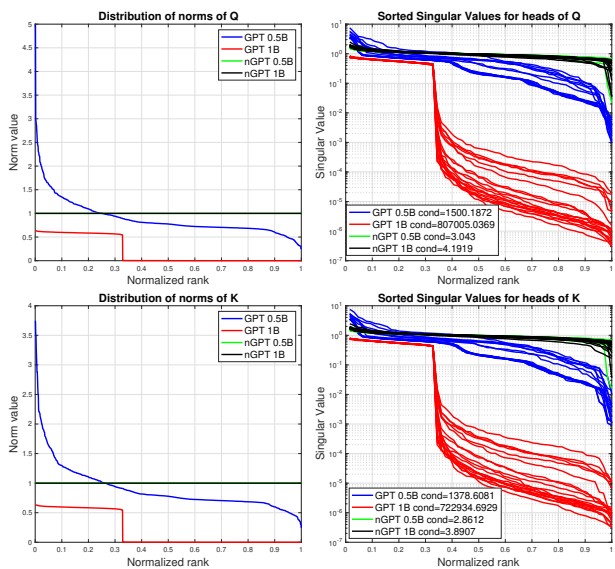

Figure 13: **Left**: Distribution of the norms of vectors forming the complete attention matrices (not per head). **Right**: Distribution of singular values for all heads. The results are shown for the 4th layer.

## A.8 LENGTH EXTRAPOLATION ABILITY

We investigate the length extrapolation ability of nGPT by evaluating its perplexity on the PG19 dataset, as shown in Figure 14. In standard GPTs, perplexity tends to increase dramatically when tested on sequences longer than pre-training length. In contrast, nGPT maintains a stable perplexity range even at extrapolated lengths. This result demonstrates that nGPT can handle longer contexts without requiring any modifications to RoPE, providing a clear advantage over standard GPTs in terms of length extrapolation.

In the original version of the paper, we explored the impact of qk-normalization on performance of nGPT. Our conclusion was that the performance is equivalent but removing qk-normalization can provide computational savings. During the review process of the paper, two reviewers asked questions which made us further investigate the role of qk-normalization and context length extrapolation abilities of nGPT. This resulted in noticing that nGPT without qk-normalization shows worse extrapolation abilities (see Figure 14-Right). This difference was initially overlooked because the results of nGPT with and without qk-normalization on other tasks are practically equivalent (see, e.g., the average accuracy on downstream tasks and validation loss of both variants in Table 6). In other words, their in-distribution performance appears the same (i.e., qk-normalization does not explain nGPT's better performance) but their out-of-distribution performance is different. The difference comes from the fact that qk-normalization normalizes $q$ and $k$, and, thus, guarantees that their dot product is bounded in [-1,1]. Even without using qk-normalization in nGPT, the hidden state and the vectors forming $W_q$ and $W_k$ matrices are normalized so that their dot product is already bounded in [-1,1]. However, the results of these individual dot products (based on vectors from $W_q$ and $W_k$) will form $q$ and $k$ per head. Moreover, $q$ and $k$ are also affected by RoPE. The qk-normalization restores $q$ and $k$ vectors back to a hyper-sphere whose dimensionality is defined by the size of each head.

While the extrapolation abilities of nGPT can be partially attested to the use of qk-normalization, the gap between nGPT without qk-normalization and the baseline GPT is still substantial. Another contributing factor to the difference is the fact that many attention matrices of the baseline GPT are practically low-rank, as discussed in the main paper. This makes them less efficient in dealing with the context. Interestingly, Kobayashi et al. (2024) recently demonstrated that weight decay induces low-rank attention layers, thus confirming our observations. While the low-rank attention matrices can potentially appear due to a set of factors, we note that weight decay is not used in nGPT.

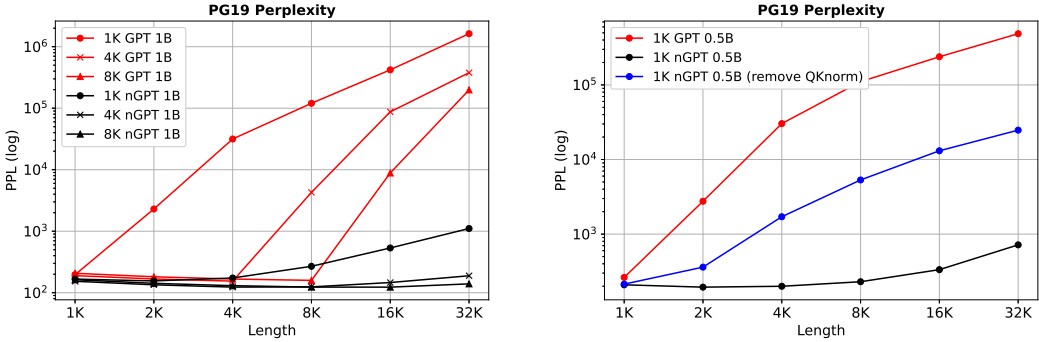

Figure 14: PG19 perplexity from 1K to 32K among different training lengths for the original GPT and nGPT (**Left**) and also with nGPT when qk-normalization is removed (**Right**).

## A.9 ABLATION STUDIES

We perform a series of ablations in nGPT, focusing on the selection of scaling factors $s_{qk}$, $s_u$ ($s_v$), $s_z$, as well as evaluating the necessity of normalization. For the scaling factors, we first examine the impact of different initializations of these parameters. Additionally, we explore whether these factors can be simplified to a learnable scalar or a fixed value. For normalization, we analyze whether the removal of QK-normalization is a feasible alternative. To ensure a fair comparison, all our ablation models have size 0.5B and context length 1k and are trained with learning rate $1 \times 10^{-3}$ for 100k iterations ($\sim$52B tokens).

Table 4 presents various combinations of $s_{init}$ and $s_{scale}$ for each scaling factor, as well as the downstream task accuracy and validation loss. Additionally, we report the mean value of each scaling factor distribution to show the final converged scaling value after training. For $s_{qk}$, we observe that the mean value (Mean($s$)) is relatievly stable with value around 1 across most initialization settings, except when using smaller $s_{init}$ and larger $s_{scale}$, which results in a Mean($s$) value less than 1. For $s_u$ and $s_v$, changes in initialization lead to significant shifts in Mean($s$), accompanied by increases in final validation loss. For $s_z$, we see a substantial degradation in both accuracy and loss under certain initializations. Overall, these results suggest that the baseline hyperparameter settings are robust but additional tuning of $s_u$, $s_v$, and $s_z$ could potentially further improve performance.

In Table 5, we modify each learnable per-element vector of the scaling factors $s_{qk}$, $s_u$ ($s_v$), $s_z$, and the eigen learning rates $\alpha_A$, $\alpha_M$ to a single learned scalar or a fixed value. This ablation helps us determine whether these tuning parameters can be simplified in nGPT. From the table, we observe that most changes result in only negligible degradation ($\leq 0.3\%$) in validation loss, and some even lead to slight improvements in accuracy. Notably, replacing the per-element vector $s_{qk}$ with a single scalar has minimal impact on the mean value, while the mean value of $s_u$, $s_v$, $s_z$, $\alpha_A$, and $\alpha_M$ show a slight increase. Furthermore, even when fixing $s_{qk}$, $s_u$, and $s_v$, the model still achieves comparable accuracy and validation loss to the baseline using the per-element vectors.

In Table 6, we investigate the impact of removing QK normalization operations and the approximation of using LERP instead of SLERP to potentially reduce training time per step. Specifically, removing the QK normalization terms (Norm) in equations 15 and 16 reduces training time per step by 12%. Conversely, replacing the baseline LERP in equation 7 with SLERP in equation 6 increases training time by 10%. Despite these changes, both modifications result in comparable accuracy and loss to the baseline, demonstrating their effectiveness in reducing computational overhead without sacrificing performance.

Table 4: Ablations of $s_{init}$ and $s_{scale}$. Our default setup is the first row of each section. **Mean($s$)** is the mean value of the learned distribution after training. **Avg. Acc** is the average accuracy of our selected five downstream tasks, **Valid. Loss** is the final validation loss, and we use $d_{model} = 1024$.

| | $s_{init}$ | $s_{scale}$ | Mean($s$) | Avg. Acc (%) ↑ | Valid. Loss ↓ |
|---|---|---|---|---|---|
| $s_{qk}$ | 1 | $1/\sqrt{d_{model}}$ | 1.51 | 54.44 | 2.252 |
| $s_{qk}$ | 0.33 | $1/\sqrt{d_{model}}$ | 1.36 | 54.67 | -0.33% |
| $s_{qk}$ | 0.05 | $1/\sqrt{d_{model}}$ | 1.01 | 53.69 | -0.09% |
| $s_{qk}$ | 1 | 1 | 1.38 | 54.19 | -0.09% |
| $s_{qk}$ | 0.33 | 1 | 1.11 | 54.89 | -0.08% |
| $s_{qk}$ | 0.05 | 1 | 0.29 | 52.41 | +1.94% |
| $s_u$ & $s_v$ | 1 | 1 | 1.24 & 1.12 | 54.44 | 2.252 |
| $s_u$ & $s_v$ | $1/\sqrt{d_{model}}$ | 1 | 0.03 & 0.03 | 53.51 | +0.92% |
| $s_u$ & $s_v$ | 1 | $1/\sqrt{d_{model}}$ | 2.75 & 11.2 | 53.90 | +0.05% |
| $s_u$ & $s_v$ | $1/\sqrt{d_{model}}$ | $1/\sqrt{d_{model}}$ | 0.40 & 0.25 | 54.39 | +0.48% |
| $s_z$ | 1 | $1/\sqrt{d_{model}}$ | 60.8 | 54.44 | 2.252 |
| $s_z$ | $\sqrt{d_{model}}$ | $1/\sqrt{d_{model}}$ | 106.1 | 52.60 | +1.06% |
| $s_z$ | 1 | 1 | 23.6 | 52.71 | +3.12% |
| $s_z$ | $\sqrt{d_{model}}$ | 1 | 63.6 | 51.84 | +1.66% |

Table 5: Ablations of replacing learnable per-element vector (scaling factors and eigen learning rate) with a single learned scalar or a fixed value. The number of the learned vector, learned scalar and fixed value are the mean values across all layers.

| $s_{qk}$ | $s_u$ & $s_v$ | $s_z$ | $\alpha_A$ & $\alpha_M$ | Avg. Acc (%) ↑ | Valid. Loss ↓ |
|---|---|---|---|---|---|
| vector = 1.47 | vector = 1.12 & 1.24 | vector = 60.80 | vector = 0.22 & 0.33 | 54.44 | 2.252 |
| **scalar = 1.49** | vector = 1.13 & 1.23 | vector = 62.11 | vector = 0.22 & 0.33 | 54.05 | +0.22% |
| vector = 1.46 | **scalar = 1.46** | vector = 60.70 | vector = 0.22 & 0.33 | 54.23 | +0.07% |
| vector = 1.47 | vector = 1.12 & 1.24 | **scalar = 95.65** | vector = 0.22 & 0.33 | 53.69 | +0.20% |
| vector = 1.40 | vector = 1.13 & 1.13 | vector = 60.90 | **scalar = 0.30 & 0.36** | 54.86 | +0.22% |
| **scalar = 1.51** | **scalar = 1.47** | vector = 61.18 | vector = 0.22 & 0.32 | 54.52 | +0.17% |
| **scalar = 1.52** | **scalar = 1.68** | **scalar = 96.65** | vector = 0.22 & 0.30 | 52.59 | +0.30% |
| **scalar = 1.49** | **scalar = 1.17** | **scalar = 88.64** | **scalar = 0.30 & 0.37** | 53.61 | +0.62% |
| **value = 1.00** | vector = 1.12 & 1.15 | vector = 60.69 | vector = 0.24 & 0.35 | 54.17 | +0.11% |
| vector = 1.45 | **value = 1.00** | vector = 61.53 | vector = 0.23 & 0.35 | 55.51 | +0.05% |
| **value = 1.00** | **value = 1.00** | vector = 61.26 | vector = 0.24 & 0.36 | 53.63 | +0.20% |
| **value = 1.00** | **value = 1.00** | **scalar = 96.15** | vector = 0.22 & 0.35 | 53.37 | +0.40% |

Table 6: Ablations of the design choices of nGPT. The removal of QK norm involves taking out the normalization terms Norm in equations 15 and 16. The replace of LERP with SLERP is using equation 6 without approximation of equation 7.

| | Training time per step (s) ↓ | Avg. Acc (%) ↑ | Valid. Loss ↓ |
|---|---|---|---|
| Baseline nGPT | 0.657 | 54.44 | 2.252 |
| 1) remove QKnorm | 0.576 | 54.71 | +0.12% |
| 2) replace LERP with SLERP | 0.726 | 54.80 | -0.08% |

## A.10 ANALYSIS OF SCALING PARAMETERS

In nGPT, we introduce a total six trainable parameters: the eigen learning rates $\boldsymbol{\alpha}_A$ and $\boldsymbol{\alpha}_M$, along with the scaling factors $\boldsymbol{s}_{qk}$, $\boldsymbol{s}_u$, $\boldsymbol{s}_v$ and $\boldsymbol{s}_z$. Since these parameters are updated using gradients, we are interested in understanding their learned distributions after training. In Figure 15, we present the histograms of these saved weights combining across layers and analyze their distributions under different context lengths, model sizes, learning rates, and numbers of training tokens.

For $\boldsymbol{\alpha}_A$ and $\boldsymbol{\alpha}_M$, we observe that their distributions remain stable across various learning rates. However, increasing the context length or the number of training tokens tends to shift the distributions to the right, indicating that the hidden state $\boldsymbol{h}$ requires more transformation from the Attention and MLP blocks to handle the larger inputs. Additionally, the mean values for these parameters tend to be smaller in larger models because we have more layers to update the hidden state.

Regarding the scaling factors, $\boldsymbol{s}_{qk}$ is quite stable under all the conditions, suggesting that a fixed value, similar to the softmax scaling factor in original transformer, may be sufficient. Interestingly, we find that $\boldsymbol{s}_{qk}$ has a high density close to zero, indicating the sparsity of attention in nGPT. For $\boldsymbol{s}_u$ and $\boldsymbol{s}_v$, their distributions shift left as the context length increases, but shift right as the model size or the number of training tokens increases. Furthermore, these two factors become flatter when using larger learning rate. Lastly, for $\boldsymbol{s}_z$, we see higher mean values for longer context lengths, larger model sizes, and more training tokens, suggesting that the model may learn to use a lower temperature to make the final word distribution sharper.

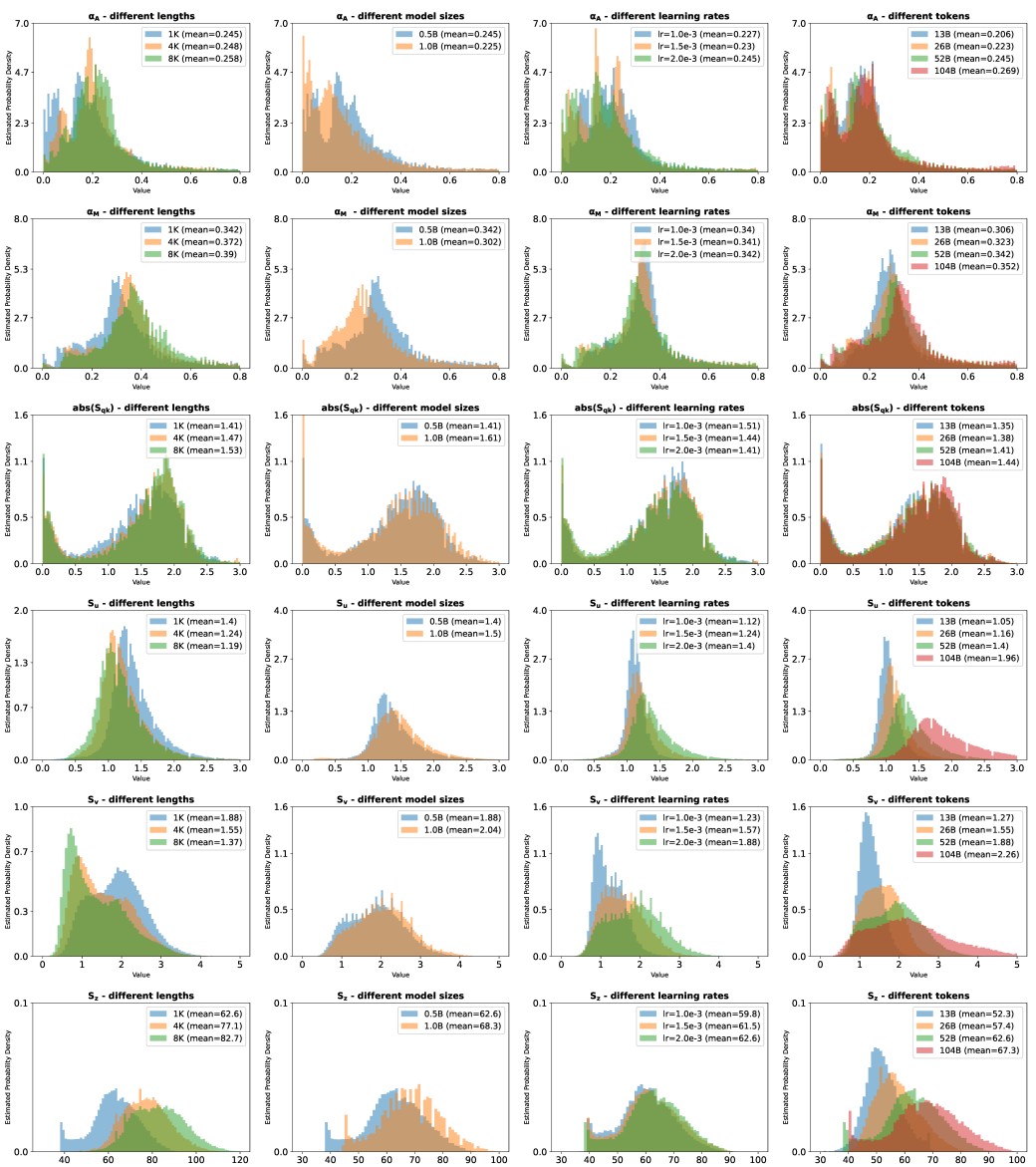

Figure 15: The distribution of trainable eigen learning rates and scaling factors varies under different context lengths, model sizes, learning rates, and number of training tokens. If not specified, we use context length 1K, model size 0.5B, learning rate $2.0 \times 10^{-3}$, and training tokens 52B as default.

