# OpenReview forum: "nGPT: Normalized Transformer with Representation Learning on the Hypersphere"
_ICLR.cc/2025/Conference — ICLR 2025 Poster_

### Official Review · Reviewer_wJPA · 2024-10-29

**Soundness:** 3
**Presentation:** 3
**Contribution:** 3
**Rating:** 6
**Confidence:** 3

**Summary:**

This paper proposed the normalized Transformer (nGPT), which normalizes all vectors forming the embeddings, MLP, attention matrices, and hidden states into unit norm to travel the tokens on the surface of a hypersphere via optimal scaling factors, trainable scaling parameters, and a variable-metric optimizer, resulting in faster convergence.

**Strengths:**

* The proposed method was well explained, and it is clear how to convert the original Transformer to the normalized Transformer.

* The correctness of each component of the proposed methods was validated through the comparison of experimental results with the original method.

* This paper proposed a variable-metric optimizer that employs LERP and eigen learning rates. This method allows the optimizer to identify an effective learning rate and discard the weight decay.

* This paper identified the appropriate scaling factor to mitigate potential risks associated with applying diverse normalization in Transformer. The proposed approach appears to improve the convergence rate of normalized Transformer.

**Weaknesses:**

**Weakness 1.** *Section 1. Introduction* provided a general overview of normalization and representation learning on the hypersphere within the context of Transformer. However, the research problem that this paper addresses and the rationale behind its significance are not sufficiently highlighted. It should be considered to emphasize these aspects before introducing the paper's contributions.

**Weakness 2.** This paper presents two novel components: one is LERP and eigen learning rates, and the other is the identification of an appropriate scaling factor. The other components, such as the normalization of all matrices and the trainable scaling factor, could be regarded as an integration of diverse normalization techniques (Weight Normalization [1], Feature Normalization [2], NormFormer [3]) and scaling parameter tuning on normalized vectors [1,4]. To further enhance the novelty of this paper, it is recommended that the novel components be validated theoretically or experimentally (e.g., experimentally validate how much the LERP approximates the SLERP after training) and that the normalized Transformer be compared with previous normalization methods, (i.e., describes the difference with other normalization techniques)

**Weakness 3.** This paper proposed numerous ablation experiments to investigate the impact of scaling factors and learnable scaling parameters on model performance. However, it seems essential to conduct an ablation study of the components of the normalized Transformer to gain a comprehensive understanding of the influence of each component on model performance.

**Questions:**

**Question 1.** Why nGPT converges faster than the original Transformer? Experimentally, the faster convergence of the normalized Transformer was confirmed, but it was not clearly explained why

**Question 2.** The rationale behind the approximation of the SLERP by LERP needs to be clarified. While the motivation and theoretical background are well explained, the purpose and reason behind this approximation still need to be clarified. Is there any reason not to use SLERP with normalization? Or, is there any benefit to using LERP instead of SLERP?

**Question 3.** The benchmarks used in the experiments solely encompass common sense reasoning and word/sentence prediction. Are the authors convinced that the experimental results are enough to indicate the versatility of the normalized Transformer in other NLP tasks? If so, I would like to know the rationale behind this conclusion.

---

**Things to improve the paper that did not impact the score**:

* [line 88] nGPT appeared before defining what it is

* This paper explained how to build it, summarized the modifications, and described hyperparameter settings in detail. Also, the normalized Transformer is easy to be implemented. However, it is recommended to publicly open the code for reproducing the paper's results.

* The structure of the paper -- which divides the proposed method into its constituent components and presents these as separate sections, and then introduces and compares the proposed method in a sequential manner -- is not a common one. This structure facilitates the reader's comprehension of the proposed method as comparing it with the baseline. However, this structure may be perceived as somewhat akin to that of a technical report. It would be advisable to restructure the paper.

* In [4], the use of a single trainable scale parameter on normalized vectors within a matrix has potential to facilitate an examination of the training behavior exhibited by the model. It may be advantageous to consider the implementation of a single trainable scale parameter for each layer or attention within the normalized Transformer. Could this method result in the generation of superior outcomes compared to those observed previously?

---

[1] Salimans, T., & Kingma, D. P. (2016). Weight normalization: A simple reparameterization to accelerate training of deep neural networks. _Advances in neural information processing systems_, _29_.

[2] Yaras, C., Wang, P., Zhu, Z., Balzano, L., & Qu, Q. (2022). Neural collapse with normalized features: A geometric analysis over the riemannian manifold. _Advances in neural information processing systems_, _35_, 11547-11560.

[3] Shleifer, S., Weston, J., & Ott, M. (2021). Normformer: Improved transformer pretraining with extra normalization. _arXiv preprint arXiv:2110.09456_

[4] Hoffer, E., Hubara, I., & Soudry, D. (2018). Fix your classifier: the marginal value of training the last weight layer. _arXiv preprint arXiv:1801.04540_.

---

> ### Author Response · Authors · 2024-11-23
> **Part 1**
>
> We thank the reviewer for the interest in our work and the valuable feedback which has greatly contributed to strengthening our paper. The segments of the revised paper that reflect the received feedback from all reviewers are marked in dark green color.
>
> **W1** *Section 1. Introduction provided a general overview of normalization and representation learning on the hypersphere within the context of Transformer. However, the research problem that this paper addresses and the rationale behind its significance are not sufficiently highlighted. It should be considered to emphasize these aspects before introducing the paper's contributions.*
>
> The original Transformer architecture is a solution in some design space which can be measured with respect to a set of metrics. Many known aspects of multi-criteria decision-making and multi-objective optimization are applied. Various transformer variants exist because depending on the selection of metrics (e.g., performance on particular tasks) and constraints (e.g., memory, compute, data), the decision-maker prefers and selects (e.g., based non-dominance or aggregation of objectives) different design solutions. The first part of the introduction describes various architectural and optimization findings/solutions of the past. Then, the authors propose to "unify various findings and observations made in the field under a new perspective of the normalized Transformer.". This unification can be viewed as a recombination procedure in the design space of architectures and optimization approaches. The introduction concludes with the description of the key aspects of the proposed solution. The experimental results of the paper demonstrate some advantages of this solution w.r.t. the baseline GPT, e.g., improved condition number of attention matrices, faster convergence.

---

> ### Author Response · Authors · 2024-11-23
> **Part 2**
>
> **W2 and Q2** *This paper presents two novel components: one is LERP and eigen learning rates, and the other is the identification of an appropriate scaling factor. The other components, such as the normalization of all matrices and the trainable scaling factor, could be regarded as an integration of diverse normalization techniques (Weight Normalization [1], Feature Normalization [2], NormFormer [3]) and scaling parameter tuning on normalized vectors [1,4]. To further enhance the novelty of this paper, it is recommended that the novel components be validated theoretically or experimentally (e.g., experimentally validate how much the LERP approximates the SLERP after training) and that the normalized Transformer be compared with previous normalization methods, (i.e., describes the difference with other normalization techniques). The rationale behind the approximation of the SLERP by LERP needs to be clarified. While the motivation and theoretical background are well explained, the purpose and reason behind this approximation still need to be clarified. Is there any reason not to use SLERP with normalization? Or, is there any benefit to using LERP instead of SLERP?*
>
> While the proposed solution can be viewed as a recombination of existing techniques, this is not how it was historically designed. The precursor of nGPT is another work by the authors (that preprint, not published in any venue, was not mentioned in the paper to avoid the risk of breaking the double-blind review), where a modification of AdamW was studied in the training of Transformer models. That work  demonstrated that instead of performing weight decay, the norm of all parameters of the network can be controlled and scheduled over the course of training. While this approach enabled acceleration in training, it was noted that the attention matrices are ill-conditioned and the normalization of the whole network or even individual matrices does not resolve it. In contrast, experiments with normalization of individual vectors within matrices demonstrated improvements of the condition numbers. Then, by combining this finding with the understanding that LayerNorm and RMSNorm normalize the hidden state to a hypersphere (when all scaling factors are equal), it was hypothesized that the spherical representation is a viable option. nGPT was designed from first principles, focusing on ensuring that the transformer architecture respects the spherical representation for all its vector components. SLERP was identified as the proper way to recombine the hidden state and the output of the transformation blocks when working on the hypersphere. Scaling factors, such as those used for logits, were introduced to relax the constraints imposed by the spherical formulation.
>
>  We have updated Appendix A.8 and Table 6 to provide a comparison between SLERP and LERP. The results of the two are very similar (we mostly focus on the validation loss difference because the average accuracy on downstream tasks is more noisy). However, SLERP is about 10\% more computationally expensive. Given that this computational overhead should become negligible for larger networks, we remain open to consider SLERP as the baseline. Our decision to select LERP for the baseline was primarily motivated by its simplicity, which, in turn, helps to simplify the presentation and adoption of nGPT. At the time of our original comparison between the two methods, we were disappointed by the small difference in performance because we expected SLERP to perform much better as a more elegant and correct method for the task.
>
>  We have updated Appendix A.2 to include a discussion of NormFormer as well as another approach called ReZero. We were also not aware of [4] by Hoffer et al. which serves as a great reference for the scaled factors of logits.

---

> ### Author Response · Authors · 2024-11-23
> **Part 3**
>
> **W3 and Q1** *This paper proposed numerous ablation experiments to investigate the impact of scaling factors and learnable scaling parameters on model performance. However, it seems essential to conduct an ablation study of the components of the normalized Transformer to gain a comprehensive understanding of the influence of each component on model performance. Why nGPT converges faster than the original Transformer? Experimentally, the faster convergence of the normalized Transformer was confirmed, but it was not clearly explained why*
>
> Understanding the individual factors contributing to the observed acceleration is central to our research interests. In addition to understanding the current approach, this would likely help to further develop it along the axes of greatest improvements. In addition to our ablation studies, we choose to highlight certain properties of the formulation (e.g., bounded dot products, group-wise effective learning rates) and acknowledge relevant observations from previous works. We believe it would be most effective to explore the acceleration factors of nGPT in a separate, comprehensive study, allowing for a thorough analysis of each contributing factor without the constraint of a page limit.
>
> **Q3** *The benchmarks used in the experiments solely encompass common sense reasoning and word/sentence prediction. Are the authors convinced that the experimental results are enough to indicate the versatility of the normalized Transformer in other NLP tasks? If so, I would like to know the rationale behind this conclusion.*
>
> This question is related to W1, where we discussed that the decision-maker selects candidate solutions based on their set of preferences and measurements performed with respect to a set of metrics. It seems highly unlikely that the results of nGPT will strictly Pareto dominate the results of the baseline GPT (given reasonable hyperparameter tuning for both) when the set of metrics of interest is very large and diverse (e.g., various NLP tasks). While the paper demonstrates that this is the case for the selected set of tasks (which are commonly used standardized tasks for evaluating language models), on some other tasks (unknown to the authors at this moment) the approach may fail. In fact, it would be of great interest to find or design such tasks to better understand the limitations of nGPT.
>
> To diversify the current selection a bit more, we have measured all trained models on a different NLP task - a machine translation task. Figure 10 shows performance on MWT14-FR-EN task which evaluates the ability (measured by the BLEU score of Papineni et al. (2002)) to translate a French sentence to English given five (French, English) example pairs, following the methodology of Radford et al. (2018) where it was used to benchmark GPT-2.
>
> **R1** *[line 88] nGPT appeared before defining what it is*
>
> Apparently, the term nGPT was initially defined only in the abstract. We have now also defined it the introduction.
>
> **R2** *This paper explained how to build it, summarized the modifications, and described hyperparameter settings in detail. Also, the normalized Transformer is easy to be implemented. However, it is recommended to publicly open the code for reproducing the paper's results.*
>
> While we are unable to publish the internal code used for the experiments, but we will provide source code built upon a popular open-source library to demonstrate the functionality of nGPT.

---

> ### Author Response · Authors · 2024-11-23
> **Part 4**
>
> **R3** *The structure of the paper -- which divides the proposed method into its constituent components and presents these as separate sections, and then introduces and compares the proposed method in a sequential manner -- is not a common one. This structure facilitates the reader's comprehension of the proposed method as comparing it with the baseline. However, this structure may be perceived as somewhat akin to that of a technical report. It would be advisable to restructure the paper.*
>
> Our intention was to introduce nGPT in a way that would be accessible even if the reader does not recall in detail how Transformers work. If the reader experiences a hesitation, it would be easier to see whether it is regarding the baseline Transformer or the introduced nGPT. We noted that other reviewers estimated the presentation of the paper as excellent, excellent and good. Given that we do not have a better idea to reorganize the presentation, we hesitate to change and potentially worsen it.
>
> **R4** *In [4], the use of a single trainable scale parameter on normalized vectors within a matrix has potential to facilitate an examination of the training behavior exhibited by the model. It may be advantageous to consider the implementation of a single trainable scale parameter for each layer or attention within the normalized Transformer. Could this method result in the generation of superior outcomes compared to those observed previously?*
>
> If we understand the recommendation correctly, the corresponding trainable scaling factors are already implemented in nGPT. We also experimented to determine whether they should be defined per parameter (e.g., in the baseline nGPT they are set per embedding parameter or per token index) or as a single scalar (as the reviewer correctly suggested/predicted). The corresponding ablation studies are discussed in Appendix A.8 and Table 5. Our conclusion was that using a single scalar factor instead of a vector leads to a moderate degradation in performance. If simplicity is the primary focus, then, indeed, most scaling factors can be just trainable scalars. While the degradation in performance is moderate, we hesitate to suggest using scalars as the default until we perform experiments on larger models to see whether the gap in performance grows.

---

> ### Comment · Reviewer_wJPA · 2024-11-25
>
> ### **Acknowledgement to the authors**
>
> I am grateful for the clarification regarding **W1** and the supplementary efforts related to **W2-3** and **Q1-3**. The revision proved instrumental in addressing the primary concern with regard to novelty and versatility.
>
>
> ### **Responses to the rebuttal**
>
> * Thanks for considering the minor thing (**R1**) and the issue of reproducibility (**R2**). I hope the author's open-source library will be publicly available.
> * The recommendation in **R3** was clearly the reviewer's personal opinion, and I agree with the author's concerns regarding the paper's restructuring. Furthermore, it was appropriate to disregard this suggestion in light of the other reviewers' evaluations of the *Presentation*.
> * Even though the explanation about **R4** has already been presented in the *Appendix* (I made a hasty judgment based only on Table 5), the authors kindly describe it again.
>
>
> ### **Additional suggestion**
>
> The issue of novelty was raised in **W2**, and the comments in **Part 2** effectively addressed it. These comments can also address **W1** by incorporating the preceding work on nGPT referenced in **Part 2** into the *Introduction* as the rationale behind exploiting the normalization instead of utilizing the weight decay. If the page limit did not permit this, it would also be beneficial to summarize the comments in the *Appendix* and refer to them in the *Introduction*.
>
>
> ### **Minor thing**
>
> [p18] According to the caption of Figure 10, the label of y-axis should be BLEU score.
>
>
> ### **Closing remarks**
> In light of the revisions above, the contribution of this paper can be distilled into two principal categories: 1) integration of existing techniques (normalization, scaling factor control) and 2) enhancement of the transformer's training strategy in terms of optimization (SLERP, eigen learning rates). The authors also demonstrated the effectiveness of the proposed method in extensive experiments.
>
> However, honestly, I am unfamiliar with this kind of paper, which primarily focuses on improving a Transformer-based model. Thus, I am still not sure that the experimental results alone support the fact that nGPT is sufficiently validated. I hope the proposed method will be validated theoretically in an extension, and this worry should not be a reason for rejecting this paper.
>
> In conclusion, I raised 1 point each of the *Representation*, *Contribution*, and *Rating* scores as the authors have addressed concerns well by revising the paper.

---

> > ### Author Response · Authors · 2024-11-28
> >
> > Thank you very much for increasing the score and for providing valuable recommendations. We have updated the paper accordingly.

---

### Official Review · Reviewer_4Taq · 2024-11-01

**Soundness:** 3
**Presentation:** 4
**Contribution:** 3
**Rating:** 6
**Confidence:** 4

**Summary:**

This paper proposes a novel architecture called the normalized Transformer (nGPT), which performs learning on a hypersphere. By normalizing parameters and output features, such as embeddings, MLPs, and attention layers, and introducing learnable scale parameters, the model enables effective representation learning on the hypersphere. The model demonstrates faster convergence compared to GPT, even without using conventional techniques like weight decay and learning rate warmup, in models with 0.5B and 1B parameters.

**Strengths:**

- Fast convergence in training is practically important for saving computational power, and this paper aims to offer a meaningful solution through the proposed architecture, nGPT.
- The model is carefully designed to prevent loss of expressiveness by introducing appropriate scaling parameters when applying existing results on the effectiveness of representation learning on the hypersphere on Transformers.
- Detailed ablation studies are conducted on the newly introduced learnable parameters, $\mathbf{s}_{qk}$, $\mathbf{s}_u$, $\mathbf{s}_v$, and $\mathbf{s}_z$, demonstrating that nGPT is robust to the introduced scale parameters.

**Weaknesses:**

- Many variations of the Transformer architecture have been proposed, but they have generally failed to successfully replace the original Transformer. This is because they have not demonstrated the scalability inherent in the original Transformer. In this paper, scalability also cannot be verified as performance has not been tested on models widely used at the 7B parameter scale or above.
- As the authors mentioned, nGPT requires an additional computation time of 80% for 4k and 60% for 8k context in a single iteration. While the authors argue that this can be reduced through code optimization, such as fused ops, the normalization of all matrices and embeddings at the beginning of each step would not benefit from fused ops, and GPU communication costs would arise with larger models. Verification of actual wall-clock time on larger models is necessary.
- The authors experimentally show that nGPT has length extrapolation ability in addition to fast convergence, but there is no discussion as to why nGPT possesses this property. Adding insights into this aspect could greatly benefit future research.

**Questions:**

- There are existing studies, such as Bachlechner et al. (2021), that demonstrate fast convergence speeds in Transformer. It would be helpful to see how nGPT compares with these previous studies.
  - Bachlechner et al., “ReZero is All You Need: Fast Convergence at Large Depth”, UAI 2021
- An interesting aspect of nGPT is the linear interpolation between the output of the attention and MLP blocks and the hidden state $h$ in Section 2.2.2. The authors used linear interpolation as shown in Equation (7), but I am curious why they did not consider directly using geodesic interpolation from Equation (6).
- It seems that the bold $\alpha$ in Equation (9) should be corrected as $\alpha$.

---

> ### Author Response · Authors · 2024-11-23
> **Part 1**
>
> We thank the reviewer for the interest in our work and the valuable feedback which has greatly contributed to strengthening our paper. The segments of the revised paper that reflect the received feedback from all reviewers are marked in dark green color.
>
> **W1** *Many variations of the Transformer architecture have been proposed, but they have generally failed to successfully replace the original Transformer. This is because they have not demonstrated the scalability inherent in the original Transformer. In this paper, scalability also cannot be verified as performance has not been tested on models widely used at the 7B parameter scale or above.*
>
> We agree with the reviewer that scalability is of great importance given that many approaches that perform well in small-scale scenarios often fail to scale effectively. Historically, this work was initially developed using 0.124B models, and in a sense (internally, for the authors), the current 1B model represents a scaled variant. Of course, this is not the scale currently in focus within the research field.
>
> Presently, we are training an 8B nGPT model and comparing it against our best in-house 8B Transformer model, which itself is a frontier-level model trained on rich, frontier-level data. Early results indicate that the 8B nGPT model achieves speedups in terms of both tokens processed and wall clock time. Based on our experience with smaller models, we anticipate that the speedup factor will increase with longer training horizons that are planned, particularly as training approaches saturation. While we may explore whether hyperparameter tuning could further enhance training, we currently use the same nGPT-specific hyperparameters as those employed for smaller models. The information provided here is not intended as a guarantee that nGPT will scale successfully with outstanding results and should not be considered as evidence supporting this submission. However, we believe it is important to share this context to emphasize that scalability is one of our primary objectives.
>
> **W2** *As the authors mentioned, nGPT requires an additional computation time of 80\% for 4k and 60\% for 8k context in a single iteration. While the authors argue that this can be reduced through code optimization, such as fused ops, the normalization of all matrices and embeddings at the beginning of each step would not benefit from fused ops, and GPU communication costs would arise with larger models. Verification of actual wall-clock time on larger models is necessary.*
>
> We agree with the reviewer that while the paper mentions that the overhead is expected to become smaller for larger models, this should indeed be verified in practice. Our current experiments with the 8B model suggest that the overhead is indeed much less significant compared to the 1B model. For instance, given that the 8B model has the same number of layers as the 1B model of the paper, the number of normalization steps does not increase (to keep the same relative overhead the number of layers would have to increase by 8). The time cost of each normalization increases linearly with the size of the embedding dimension, which, in turn, increases only sub-linearly when scaling from 1B to 8B. Since most of the increase of the per step computational complexity is concentrated in matrix-vector multiplications (unaffected by nGPT), the relative overhead of nGPT-related operations is greatly diminished. This information cannot be counted towards this submission, nevertheless, we think that it might be useful to share it.

---

> ### Author Response · Authors · 2024-11-23
> **Part 2**
>
> **W3** *The authors experimentally show that nGPT has length extrapolation ability in addition to fast convergence, but there is no discussion as to why nGPT possesses this property. Adding insights into this aspect could greatly benefit future research.*
>
> The reviewer's suggestion to investigate why nGPT demonstrates better context length extrapolation abilities (which is also linked to the reviewer Z8Gb suggestion to clarify if we can omit the normalization of q and k) prompted us to take a closer look at this question. This investigation revealed that the normalization of q and k has a significant impact on this ability. Please consider the following results  of 0.5B models trained with 1k context length and measured on 8k and 32k context variants of PG19 (see also Figure 13-Right for 4k and 8k training context length):
>
> \# baseline gpt
>
> pg19-seqlen-1024                  262.9
>
> pg19-seqlen-4096              	30298.7862
>
> pg19-seqlen-16384             	238384.2578
>
> \# baseline ngpt
>
> pg19-seqlen-1024                  209.1
>
> pg19-seqlen-4096              	199.9253
>
> pg19-seqlen-16384             	333.8669
>
> \# baseline ngpt without qknorm
>
> pg19-seqlen-1024                  213.6
>
> pg19-seqlen-4096              	1709.5132
>
> pg19-seqlen-16384             	13095.9196
>
> The results suggest that nGPT without qk-normalization shows worse extrapolation abilities. This difference was initially overlooked because the results of nGPT with and without qk-normalization on other tasks are practically equivalent (see, e.g., the average accuracy on downstream tasks and validation loss of both variants in Table 6). In other words, their in-distribution performance appears the same (i.e., qk-normalization does not explain nGPT's better performance) but their out-of-distribution performance is different. The difference comes from the fact that qk-normalization normalizes q and k, and, thus, guarantees that their dot product is bounded in [-1,1]. Even without using qk-normalization in nGPT, the hidden state and the vectors forming Wq and Wk matrices are normalized so that their dot product is already bounded in [-1,1]. However, the results of these individual dot products (based on vectors from Wq and Wk) will form q and k per head. Moreover, q and k are also affected by RoPE. The qk-normalization restores q and k vectors back to a hyper-sphere whose dimensionality is defined by the size of each head.
>
> While the extrapolation abilities of nGPT can be partially attested to the use of qk-normalization, the gap (see, e.g., the results for pg19-seqlen-4096 above)  between nGPT without qk-normalization and the baseline GPT is still substantial. Another contributing factor to the difference is the fact that many attention matrices of the baseline GPT are practically low-rank as discussed in the main paper. This makes them less efficient in dealing with the context.
> Interestingly, the recently appeared NeurIPS 2024 + arXiv paper titled "Weight decay induces low-rank attention layers" by S. Kobayashi et. al. (this research is independent from ours) confirms our observations. While the low-rank attention matrices can potentially appear due to a set of factors, we note that weight decay is not used in nGPT.
>
> In addition to these contributing factors specific to the attention block, it is likely that the better performance of nGPT within the training context length should also contribute to its ability during extrapolation when prediction errors accumulate.
>
> We have updated the text of the paper to clarify the role of qk-normalization.

---

> ### Author Response · Authors · 2024-11-23
> **Part 3**
>
> **Q1** *There are existing studies, such as Bachlechner et al. (2021), that demonstrate fast convergence speeds in Transformer. It would be helpful to see how nGPT compares with these previous studies. Bachlechner et al., “ReZero is All You Need: Fast Convergence at Large Depth”, UAI 2021*
>
> We were not aware of the study of Bachlechner et al. mentioned by the reviewer.
> We have updated our discussion of eigen learning rates in the Appendix A.2 to analyze how the update equation proposed by Bachlechner et al. is related to the update equation of nGPT.
>
> **Q2** *An interesting aspect of nGPT is the linear interpolation between the output of the attention and MLP blocks and the hidden state in Section 2.2.2. The authors used linear interpolation as shown in Equation (7), but I am curious why they did not consider directly using geodesic interpolation from Equation (6).*
>
> We have updated Appendix A.8 and Table 6 to provide a comparison between SLERP and LERP. The results of the two are very similar (we mostly focus on the validation loss difference because the average accuracy on downstream tasks is quite noisy). However, SLERP is about 10\% computationally more expensive. Given that the latter computational overhead should become negligible for larger networks, we remain open to consider SLERP as the baseline. Our decision to select LERP for the baseline was primarily motivated by its simplicity, which, in turn, could help to simplify the presentation and adoption of nGPT. At the time of our original comparison of the two methods, we were disappointed by the small difference in performance because we expected SLERP to perform much better as a more elegant and correct method for the task.
>
> **Q3** *It seems that the bold alpha in Equation (9) should be corrected as alpha*
>
> We overlooked that unfortunately. We have fixed that typo.

---

> > ### Comment · Reviewer_4Taq · 2024-11-25
> >
> > I sincerely appreciate the authors' responses. The experiments on the impact of qk-normalization on extrapolation performance are highly informative, and the discussion related to weight decay is insightful. Additionally, the inclusion of comparisons with prior works aimed at improving convergence further enhances the quality of the paper. The comparison experiments between SLERP and LERP also provide valuable information for other researchers.
> >
> > However, the practical value of the proposed nGPT from a real-world application perspective has not yet been fully demonstrated. Despite this, I believe this paper deserves a positive rating, and I will maintain my original positive rating. I hope future work will include further validation of nGPT's scalability, ensuring it becomes a widely adopted architecture.

---

> > > ### Author Response · Authors · 2024-11-28
> > >
> > > Thank you very much for maintaining the positive rating for our work.

---

### Official Review · Reviewer_MiZy · 2024-11-02

**Soundness:** 3
**Presentation:** 3
**Contribution:** 3
**Rating:** 6
**Confidence:** 3

**Summary:**

The paper introduces a novel neural network architecture, the normalized Transformer (nGPT) with representation learning on the hypersphere, and empirically shows that nGPT learns significantly faster, cutting down the training steps needed to reach comparable accuracy levels.

**Strengths:**

- The paper is well-written, and the related work section is presented clearly.

- The experimental results are solid and convincing.

**Weaknesses:**

- I do not find any major weaknesses. My only concern is that, in essence, the method involves normalizing everything to unit, functioning as an embedding layer within architectures, with empirical results demonstrating its effectiveness. Although hypersphere normalization in representation learning is introduced and applied, the paper provides a clear discussion of related papers.

**Questions:**

I have no question.

---

> ### Author Response · Authors · 2024-11-23
> **Our reply**
>
> We thank the reviewer for their positive evaluation of our work and their support.

---

> > ### Comment · Reviewer_MiZy · 2024-11-26
> >
> > Thank you for the response. The score is maintained.

---

> > > ### Author Response · Authors · 2024-11-28
> > >
> > > Thank you very much for maintaining the positive rating for our work.

---

### Official Review · Reviewer_Z8Gb · 2024-11-04

**Soundness:** 3
**Presentation:** 4
**Contribution:** 4
**Rating:** 8
**Confidence:** 3

**Summary:**

Motivated by previous findings on the advantages of representation learning on a hypersphere, the authors propose a novel neural network architecture called the normalized Transformer (nGPT). In nGPT, all vectors—including embeddings, MLP outputs, attention matrices, and hidden states—are normalized to ensure they reside on the same hypersphere. This setup ensures that each token in the input sequence, during optimization, starts at a position on the hypersphere defined by its input embedding and moves on the surface of the hypersphere to a point that best predicts the embedding of the next token. Experiments on the OpenWebText dataset demonstrate that nGPT learns significantly faster, reducing the number of training steps needed to achieve performance comparable to baseline models.

**Strengths:**

The paper is well-written and well-structured. The pair-wise comparisons in Section 2 make it easy to understand the architectural modifications, with clear motivation and explanations for each change.

The comprehensive experiments and ablation studies effectively demonstrate the method's effectiveness. The experimental results are impressive.

**Weaknesses:**

Minor: The per-step time of nGPT is approximately 80% higher for a 4k context and 60% higher for an 8k context, primarily due to additional normalization steps. However, Table 6 shows that removing the normalization for q and k only slightly affects performance, suggesting a potential way to reduce overhead.

**Questions:**

**Q1:** The authors comment on the average magnitudes of $\alpha_A$ and $\alpha_M$, noting that $\alpha_M$ has a higher average magnitude than $\alpha_A$, implying their suggestions more precise. could the authors the authors elaborate on this point in more detail?

**Q2:** Previous research has demonstrated that representation learning on a hypersphere enhances training stability, with experiments and ablations confirming stable activations, gradients, and weights, as well as showing gradient descent acting as a meta-optimizer. The 4-20x speedup achieved by the normalized Transformer is impressive, and further intuition or analysis on the factors contributing to this acceleration would be valuable.

---

> ### Author Response · Authors · 2024-11-23
> **Part 1**
>
> We thank the reviewer for the interest in our work and the valuable feedback which has greatly contributed to strengthening our paper. The segments of the revised paper that reflect the received feedback from all reviewers are marked in dark green color.
>
> **Q1** *The authors comment on the average magnitudes of alphaA and alphaM, noting that
>  alphaM has a higher average magnitude than alphaA, implying their suggestions more precise. could the authors elaborate on this point in more detail?*
>
> The introduced alphaA and alphaM parameters definitely require an individual study to better understand their role and impact. In their current formulations, these parameters rescale the step taken towards the output of the transformation block. Our current understanding is that the scale of these parameters is likely to be linked to the predictive power of the transformation block. For instance, our prediction would be that if the block has just a few parameters (e.g., a very small MLP), then its potential maximum predictive capacity would be lower than for a block of a much greater size (e.g., a much larger MLP which can potentially contain the smaller MLP as its particular case). If the larger block provides a better prediction of how to update the hidden state, then the optimization procedure (e.g., Adam) is likely to optimize it to have a larger eigen learning rate. Conversely, if a very small MLP block provides predictions that are not much better than random, then the optimization procedure could limit its impact on the hidden state by lowering the corresponding eigen learning rates.
>
> In the paper, we mentioned "alphaM decreases from 0.37 to 0.32, possibly because MLP blocks have more parameters, making their suggestions more precise." to reflect on our hypothesis described above: parameters count is likely linked to maximum possible predictive power of the corresponding transformation block. Thus, the blocks with more parameters (in our experimental settings, MLP blocks have more parameters than Attention blocks) are likely to have higher maximum possible predictive power and the corresponding eigen learning rates are likely to be higher for them.
>
> It should be noted that the view and predictions described above are expected to better fit observations at the late stage of optimization. During the earlier stages of training, the optimization can potentially implement some form of curriculum learning by lowering eigen learning rates of some blocks in order to effectively search in a space of smaller architectures (e.g., when contributions of some layers are negligible). This research direction is of great interest on its own and requires a focused approach with targeted experiments and ablations. We have updated the text of the corresponding text block in Section 3.2 to better clarify our view.
>
> We have also updated Appendix A.2 to answer a related question of reviewer 4Taq.

---

> ### Author Response · Authors · 2024-11-23
> **Part 2**
>
> **W1** *Minor: The per-step time of nGPT is approximately 80\% higher for a 4k context and 60\% higher for an 8k context, primarily due to additional normalization steps. However, Table 6 shows that removing the normalization for q and k only slightly affects performance, suggesting a potential way to reduce overhead.*
>
> The reviewer is absolutely correct in noticing that removing the normalization for q and k only slightly affects performance, making it feasible to remove the normalization to reduce overhead. However, this question and also the question of reviewer 4Taq regarding extrapolation abilities of nGPT made us investigate it further. We found that qk-normalization is beneficial when scaling up the trained model to larger context lengths. In order to avoid duplicating our reply, the detailed explanation is given within the section of reviewer 4Taq.
> We have updated the text to highlight our new observations. Additionally, we now explicitly mention the potential benefit of removing qk-normalization:  approximately a 12\% reduction in the computational time per step. On the other hand, we also mention the benefits of qk-normalization when the context length extrapolation is of importance.
>
> **Q2** *Previous research has demonstrated that representation learning on a hypersphere enhances training stability, with experiments and ablations confirming stable activations, gradients, and weights, as well as showing gradient descent acting as a meta-optimizer. The 4-20x speedup achieved by the normalized Transformer is impressive, and further intuition or analysis on the factors contributing to this acceleration would be valuable.*
>
> Understanding the individual factors contributing to the observed acceleration is central to our research interests. In addition to understanding the current approach, this would likely help to further develop it along the axes of greatest improvements. In addition to our ablation studies, we choose to highlight certain properties of the formulation (e.g., bounded dot products, group-wise effective learning rates) and acknowledge relevant observations from previous works. We believe it would be most effective to explore the acceleration factors of nGPT in a separate, comprehensive study, allowing for a thorough analysis of each contributing factor without the constraint of a page limit.

---

> ### Comment · Reviewer_Z8Gb · 2024-11-26
>
> Thank you for the rebuttal. My concerns have been addressed, and I remain positive in my vote for the paper.

---

> > ### Author Response · Authors · 2024-11-28
> >
> > Thank you very much for maintaining the positive rating for our work.

---

### Author Response · Authors · 2024-11-23
**Response to All Reviewers**

We thank all reviewers for their positive evaluation and the valuable feedback which has greatly contributed to strengthening our paper. We have uploaded a revision to address the issues raised and individually reply to the reviewers' concerns. The segments of the revised paper that reflect the received feedback from all reviewers are marked in dark green color. Modifications include i) a comparison of SLERP and LERP, ii) an analysis of impact of qk-normalization on context length extrapolation, iii) an analysis of related works mentioned by reviewers, and iv) Figure 10 with results on a machine translation task.

We kindly ask you to update your rating if our replies have addressed your concerns.

Thank you again for your reviews!

---

### Public Comment · ~Weiyang_Liu1 · 2024-11-26
**Some related papers**

Congrats on the great work. It is good to see representation learning on hypersphere with the transformer architecture.   :)

We have a few works that are highly relevant to nGPT. Maybe they are of sufficient interest to the authors.

[1] Deep Hyperspherical Learning, NIPS 2017 (https://arxiv.org/abs/1711.03189)

[2] Decoupled Networks, CVPR 2018 (arXiv preprint arXiv:1804.08071)

---

> ### Author Response · Authors · 2024-11-28
>
> Thank you very much for the relevant references. We have added them to the text regarding key findings and observations made in the field which directly support representation learning on the hypersphere.

---

### Public Comment · ~Yuan_Yao14 · 2025-04-20
**What is the affect of normalized v in the Attention modular?**

Dear Authors,

In my opinion, nGPT is elegant and represents a significant advancement in the field. **I noticed that the vector v in the Attention modular is not normalized in your model**. Could you please share whether you have conducted any experiments involving the normalization of vector v? If so, did those experiments lead to any performance degradation or other notable outcomes?

Thank you for your time and consideration.

Best regards,

Yuan

---

### Meta-Review · Area_Chair_h9Df · 2024-12-20

**Metareview:**

This work proposes modifying the standard transformer architecture to normalize various aspects of the architecture (weight vectors, embeddings, etc) to lie on the unit sphere.  The authors note that this allows for the matrix-vector multiplications to be interpreted as cosine-similarity dot products and show empirically that the resulting architecture trains in significantly fewer optimization iterations than standard transformers.

The reviewers are somewhat borderline to positive. One primary concern raised by the reviewers is the scalability of the proposed method, as the computation per iteration is significantly larger than that of a standard transformer.  The authors note that this increased computation per iteration is offset by the smaller number of iterations needed to train the network and the potential for further optimizations in the implementation of the approach.

Overall, the idea is fairly straight-forward and borrows from similar ideas in other (i.e., non-transformer) architectures.  Nevertheless, while a relatively simple idea, I believe the experimental results will be of interest to the community both for practitioners as well as theoreticians exploring topics such as optimization dynamics.

**Additional Comments On Reviewer Discussion:**

The authors were largely responsive to the reviewers' comments, with some reviewers increasing their scores.

---

### Decision · Program_Chairs · 2025-01-22

Accept (Poster)